🔓 | **Open Peer Review** | Environmental Microbiology | Research Article

# Deciphering the biosynthetic landscape of biofilms in glacier-fed streams

Aileen Ute Geers,[1] Grégoire Michoud,[1] Susheel Bhanu Busi,[2] Hannes Peter,[1] Tyler J. Kohler,[1,3] Leïla Ezzat,[1,4] The Vanishing Glaciers Field Team,[1] Tom J. Battin[1]

**ABSTRACT** Glacier-fed streams are permanently cold, ultra-oligotrophic, and physically unstable environments, yet microbial life thrives in benthic biofilm communities. Within biofilms, microorganisms rely on secondary metabolites for communication and competition. However, the diversity and genetic potential of secondary metabolites in glacier-fed stream biofilms remain poorly understood. In this study, we present the first large-scale exploration of biosynthetic gene clusters (BGCs) from benthic glacier-fed stream biofilms sampled by the *Vanishing Glaciers* project from the world's major mountain ranges. We found a remarkable diversity of BGCs, with more than 8,000 of them identified within 2,868 prokaryotic metagenome-assembled genomes, some of them potentially conferring ecological advantages, such as UV protection and quorum sensing. The BGCs were distinct from those sourced from other aquatic microbiomes, with over 40% of them being novel. The glacier-fed stream BGCs exhibited the highest similarity to BGCs from glacier microbiomes. BGC composition displayed geographic patterns and correlated with prokaryotic alpha diversity. We also found that BGC diversity was positively associated with benthic chlorophyll *a* and prokaryotic diversity, indicative of more biotic interactions in more extensive biofilms. Our study provides new insights into a hitherto poorly explored microbial ecosystem, which is now changing at a rapid pace as glaciers are shrinking due to climate change.

**IMPORTANCE** Glacier-fed streams are characterized by low temperatures, high turbidity, and high flow. They host a unique microbiome within biofilms, which form the foundation of the food web and contribute significantly to biogeochemical cycles. Our investigation into secondary metabolites, which likely play an important role in these complex ecosystems, found a unique genetic potential distinct from other aquatic environments. We found the potential to synthesize several secondary metabolites, which may confer ecological advantages, such as UV protection and quorum sensing. This biosynthetic diversity was positively associated with the abundance and complexity of the microbial community, as well as concentrations of chlorophyll *a*. In the face of climate change, our study offers new insights into a vanishing ecosystem.

**KEYWORDS** secondary metabolites, microbiomes, glacier-fed streams, biofilms

Biofilms consist of complex assemblages of bacteria, archaea, microeukaryotes, and viruses, all enclosed within an extracellular matrix (1, 2). In streams and rivers, microorganisms predominantly live within biofilms, which play crucial roles in ecosystem processes, including metabolism, and carbon and nutrient cycling (3, 4). Biofilms also form the foundation of the riverine food web, thereby influencing higher trophic levels (3). This is particularly true for glacier-fed streams, where biofilms are the dominant life form. Despite harsh environmental conditions (e.g., low temperature, ultra-oligotrophy) in glacier-fed streams, biofilms include a diverse microbiome (5). Mixotrophy, diverse

**Peer Reviewer** Hannah Doris Schweitzer, UiT-The Arctic University of Norway, Tromsø, Norway

Address correspondence to Tom J. Battin, tom.battin@epfl.ch.

The authors declare no conflict of interest.

See the funding table on p. 18.

energy acquisition pathways, internal nutrient recycling, and cross-domain interactions are thought to contribute to the success of the biofilm mode of life in glacier-fed streams (6, 7). Biotic interactions between photoautotrophs and microbial heterotrophs were recently predicted to increase in future glacier-fed streams because of glacier shrinkage (7, 8). This is due to more favorable environmental conditions in glacier-fed streams, promoting primary production and likely the development of more copious and complex biofilms (8).

The division of labor in biofilms, involving the production and distribution of public goods, exploitation of resources, communication, and competition are all facilitated by secondary metabolites (9–12). Often also referred to as "specialized metabolites," secondary metabolites are characterized by their specificity and distinct functions in contrast to the various main metabolic products; they are not directly involved in growth and reproduction (13, 14). Secondary metabolites can be classified into five broad categories based on chemical structure and biosynthesis: (i) non-ribosomal peptides (NRP), (ii) polyketides (PKs), (iii) ribosomally synthesized and post-translationally modified peptides (RIPPs), (iv) terpenes, and (v) hybrids of the previous four. However, there are many more secondary metabolites that do not fit into these categories or have not yet been characterized (15). Functional roles of secondary metabolites in the microbiome include (but are not limited to) competition, communication, metal sequestration, antioxidants, regulation of membrane fluidity, motility, and formation of biofilms (10, 16–18). Secondary metabolites that offer an adaptive advantage to extreme environmental conditions and the biofilm mode of life may be particularly relevant in glacier-fed streams. This includes secondary metabolites involved in the uptake of iron (e.g., siderophores, vibrioferrin [19]), UV radiation and oxidative stress protection (e.g., pigments, arylpolyenes [20–22]), osmotic and temperature stress protection (e.g., ectoine [23, 24]), as well as in biofilm formation and quorum sensing (e.g., homoserine lactones [25–27], argD-like lacton autoinducers [28, 29]).

Genes responsible for the synthesis of secondary metabolites are frequently clustered together in bacterial genomes, forming biosynthetic gene clusters (BGCs) (30). Using this characteristic clustering to detect biosynthetic genes for secondary metabolites in metagenomic sequences has unveiled a vast and unexplored reservoir of BGCs within environmental microbiomes (31). For example, in the global ocean microbiome, an excess of 60,000 BGCs were identified, with their diversity structured according to temperature variation and sample depth (32). In contrast, a large-scale analysis of soil microbiomes suggested that geographical distance predominantly governs biosynthetic diversity rather than environmental characteristics (33, 34). However, the study of secondary metabolites is complicated by the fact that some secondary metabolites can be synthesized by several BGCs, which share sequence similarity but are not identical. To overcome this challenge, BGCs can be organized into gene cluster families (GCFs) with the assumption that BGCs within a gene cluster family synthesize the same secondary metabolite. Current approaches (e.g., Big-SCAPE [35], BiG-SLiCE [36]) use sequence similarity and the order of genes within BGCs to group them into GCFs. These methods can also be used to differentiate novel BGCs from known BGCs in databases such as NCBI.

While the role and diversity of secondary metabolites have been investigated in marine and terrestrial ecosystems (32, 37), little is known about their diversity and potential importance in glacier-fed streams. A previous study has indicated diverse secondary metabolites, potentially related to anti-bacterial functions, in glacier-fed stream biofilms in the New Zealand Southern Alps and in the Caucasus (38). Work on glaciers (39) and Antarctic soils (40) suggests a large degree of novelty and heterogeneity of BGCs in cryospheric ecosystems. Diverse pigments in cold-adapted bacteria (41) and large proportions of terpene BGCs in glacier (39) and marine (32) microbiomes suggest a role in the protection against UV radiation and contribution to photosynthesis.

The aim of the present study was to characterize the genetic potential of the glacier-fed stream microbiome for secondary metabolites based on BGCs in metagenome-assembled genomes (MAGs) from benthic biofilms. Biofilms were sampled by the

*Vanishing Glaciers* project from 85 glacier-fed streams across the European Alps, Caucasus Mountains, Ecuadorian Andes, Southwest Greenland, Pamir and Tien Shan, Himalayas, Southern Alps, Rwenzori, and Scandinavian Mountains. To capture the heterogeneity inherent to stream ecosystems, partially imposed by various sedimentary habitats, we sampled both epipsammic (i.e., attached to sandy sediments) and epilithic (i.e., attached to boulders) biofilms. We also sampled each glacier-fed stream from an upstream reach close to the glacier snout and a downstream reach close to the terminal moraine of the Little Ice Age. To couple biosynthetic diversity to prokaryotic and photosynthetic eukaryotic taxonomic diversity, we performed amplicon sequencing of the 16S rRNA and 18S rRNA genes. We anticipated to discover a diverse repertoire of secondary metabolite synthesis capabilities distinct to other environmental microbiomes. We hypothesized that elevated prokaryotic alpha diversity and benthic algal biomass are associated with higher biosynthetic potential for secondary metabolites.

## RESULTS

### Diverse BGC composition within glacier-fed stream microbiomes

To explore the potential of the glacier-fed stream microbiome for secondary metabolite biosynthesis, we analyzed 2,855 bacterial and 13 archaeal medium-to-high quality MAGs. MAGs were assembled using 173 shotgun metagenomic samples from both epipsammic and epilithic biofilms in the up- and downstream reaches of the 85 glacier-fed streams (7). We identified 8,040 BGCs using AntiSMASH, which capitalizes on the inherent clustering of BGCs and the similarity of signature proteins to detect potentially novel BGCs. We subsequently organized the identified BGCs into 6,184 gene cluster families (GCFs) based on sequence similarity using BiG-SCAPE (35) (Table S1). Approximately 11% of the BGCs were encoded within the terminal ends of contigs, suggesting their likely completeness; 12.5% of the GCFs had at least one complete BGC. The average size of the BGCs was 13.9 ± 8.9 kbp, with a maximum of 98 kbp. Eight BGCs were found on archaeal MAGs, predominantly affiliated with the phylum Halobacteriota, and categorized as either ribosomally synthesized and post-translationally modified peptides (RIPP) or terpene BGCs. The remaining 8,032 BGCs were from bacterial MAGs, with terpenes (29.3%) and RiPPs (23.7%) constituting the majority, followed by polyketide synthases (PKSs; 14.27%), non-ribosomal peptide synthetases (NRPSs; 13.4%), PKS–NPRS hybrids (2.0%), and other categories (17.3%).

The average number of BGCs per MAG was 2.8 ± 2.8, encompassing a range from 0 BGCs per MAG up to a maximum of 28 BGCs per MAG. The phyla encoding the highest potential to produce secondary metabolites were Eremiobacterota, Fibrobacterota, Elusimicrobiota, and Myxococcota, reaching 8.7 ± 5.5 BGCs per MAG for Myxococcota (Fig. 1A). The high number of BGCs per MAG in the phylum Myxococcota was mainly driven by the families *Polyangiaceae* (10.7 ± 5.4), *Myxococcaceae* (8.8 ± 6.7), and *Haliangiaceae* (5.7 ± 2.8). The distribution of different BGC categories was comparable across phyla, dominated by terpenes and RiPPs. Myxococcota were at the extreme end of the distribution, with the second lowest proportion of terpenes (15.4%), while only Desulfobacterota displayed a lower percentage of terpenes (12.8%; Fig. 1B, Table S2). An elevated proportion of BGCs not belonging to the five main categories was observed in the Desulfobacterota. These "other" BGCs in Desulfobacterota were predominantly arylpolyenes (48% of "other" BGCs) and resorcinol (Table S2). In contrast, Eremiobacterota demonstrated a notable prevalence of terpene BGCs (54.5%), and Deinococcota had the highest proportion at 45.0% of RiPPs (Fig. 1B).

To assess the novelty of BGCs in the glacier-fed stream microbiome, we compared them to the NCBI reference database, which includes 1.2 million BGCs (36). We used BiG-SLiCE to extract Pfam domains, which are then used to compute Euclidean distances of all glacier-fed stream BGCs to the NCBI BGCs (Table S1). Employing the standard threshold of Euclidean distance greater than 900 (36), we found that 40.5% of the glacier-fed stream BGCs are novel, meaning that they are not part of a known GCF and thus likely produce unknown secondary metabolites. An average of 859.6 ± 313.8 Euclidean

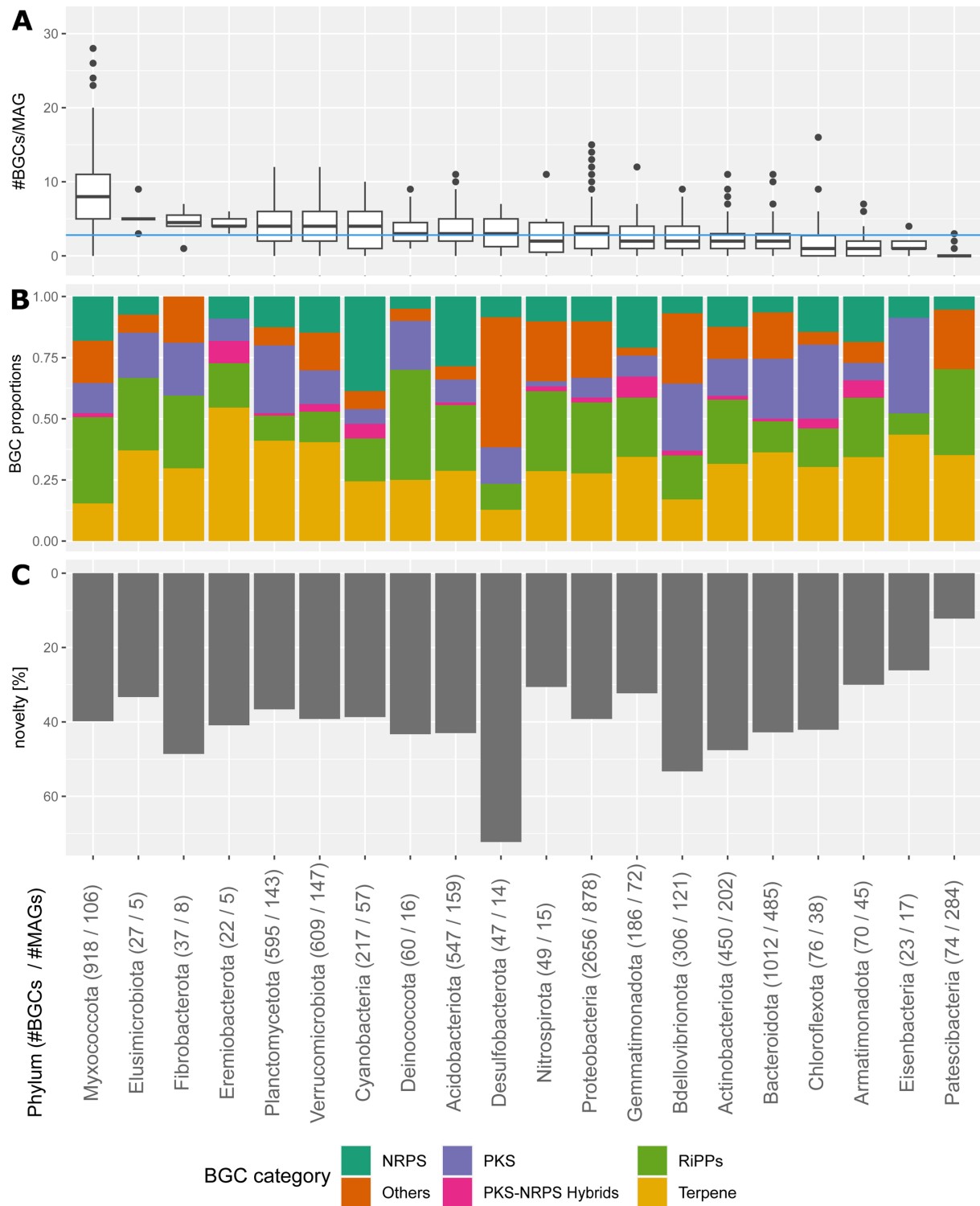

**FIG 1** The biosynthetic potential in glacier-fed streams assessed based on the number of BGCs per MAG, BGC category composition, and BGC novelty in the most abundant bacterial phyla. On the x-axis are the different phyla (BGC number/MAG number) ordered by their average number of BGCs per genome. Phyla with less than 10 BGCs are excluded. The x-axis is the same for all subfigures A–C. (A) The number of BGCs per MAG. The blue horizontal line denotes the average number of BGCs across all MAGs across all phyla. (B) Relative prevalence of BGC categories: non-ribosomal peptide synthetase (NRPS), polyketide synthase (PKS), ribosomally synthesized and post-translationally modified peptides (RIPPs), PKS–NPRS hybrids, terpenes, and others (C) Percentage of novel BGCs. Novelty is determined by a comparison to known BGCs sourced from the NCBI database using BiG-SLiCE. The standard threshold of a Euclidian distance (based on identified Pfam domains) >900 was taken.

distance to known BGCs highlights the unique array of biosynthesis of secondary metabolites. Remarkably, 37 BGCs exhibited Euclidean distances exceeding 1,800, indicating substantial genetic divergence from previously sequenced BGCs and potentially novel biosynthetic pathways. The phylum Desulfobacterota displayed the highest percentage of novel BGCs at 72.3%, followed by Bdellovibrionota (53.3%), Fibrobacterota (48.6%), and Actinobacteriota (47.6%) (Fig. 1C).

## The biosynthetic potential differs between glacier-fed stream and other aquatic microbiomes

We compared the BGCs from glacier-fed streams with those found in other aquatic microbiomes (e.g., Tibetan glaciers [39] and rivers [42], Tibetan and Canadian freshwater lakes [42, 43], Tibetan wetlands [42], and global pelagic ocean [32]; Table S3). The ratio of BGCs to MAGs was highest in the Tibetan glacier microbiome (3.93), followed by global glacier-fed streams (2.80), Tibetan freshwater lakes (2.48), wetlands (2.12) and rivers (1.99), Canadian freshwater lakes (1.83), and the global pelagic ocean (1.22). Additionally we grouped all BGCs into 20,640 GCFs, by applying the default threshold of BiG-SLiCE (36). While terpene GCFs were elevated in Canadian freshwater lakes (45.6%), Tibetan freshwater lakes (34.6%), and rivers (36.3%), RiPP GCFs were more prevalent in glacier-fed streams (24.7%), Tibetan glaciers (22.0%), and wetlands (21.7%). The global ocean microbiome had the highest proportion of "other" GCFs (27.6% Fig. 2A).

We assessed the biosynthetic overlap between microbiomes by investigating which GCFs were present in one, two, or more microbiomes (Fig. 2B). The majority of GCFs were present in only one microbiome. When we examined the microbiomes individually, we saw that the GCF uniqueness (i.e., GCFs found in only one microbiome) ranged from 28% in Tibetan rivers to 76% in the pelagic ocean. In glacier-fed streams, this percentage of uniqueness was in the upper range at 62%. The highest overlap was observed between Tibetan rivers, lakes, and wetlands (Fig. 2B). To better quantify how many GCFs, and thus likely secondary metabolites, are shared between microbiomes, we calculated the Jaccard distances based on GCF presence and absence. Glacier-fed streams displayed the closest Jaccard distances to the Tibetan glaciers (0.89), followed by Tibetan wetlands (0.90, Fig. S1A). Only 111 GCFs were present in all microbiomes (Fig. 2B), with 39.6% of them being terpenes, which is higher than the average percentage of terpenes across all microbiomes (34.2%). In contrast, 8.1% of the ubiquitous GCFs were RiPPs, which is lower than the average (20.1%) across microbiomes. Notably, there was a significant positive correlation of the size of GCFs (i.e., the number of BGCs per GCF) with their prevalence (i.e., the number of microbiomes a GCF can be found in) (Fig. S1B).

## BGC diversity varies between biofilm types, mountain ranges, and glacier influence

We determined the abundance of glacier-fed stream GCFs by mapping the metagenomic reads of 173 samples (i.e., including up- and downstream reaches) (Table S4) to a set of representative BGCs for each GCF. We also explored differences in GCF abundances between epilithic and epipsammic biofilms, which have been reported to harbor differing microbiomes (44). Abundances of the GCFs were normalized by the abundance of the *recA* gene, a proxy for cell abundance, to account for sequencing depth. We found higher average Shannon alpha diversity of GCFs in epipsammic (6.9 ± 0.3) than epilithic (5.4 ± 1.2) biofilms (Student's *t*-test $P = 8.26 \times 10^{-5}$). In contrast, the GCF alpha diversity did not significantly vary between mountain ranges in the epipsammic biofilms (ANOVA, $P = 0.522$; Fig. 3A). Additionally, GCF alpha diversity was significantly correlated with bacterial abundance, determined using flow cytometry, in the epipsammic biofilms (Pearson's r = 0.26, $P$ adj = 0.029), but not with benthic chlorophyll *a*, a proxy for algal biomass (Fig. 3B).

Previous studies have indicated a strong dependence of biosynthetic diversity on the taxonomic diversity of the microbiome and significant vertical inheritance of BGCs (45–47). In order to further link prokaryotic, photosynthetic eukaryotic, and biosynthetic

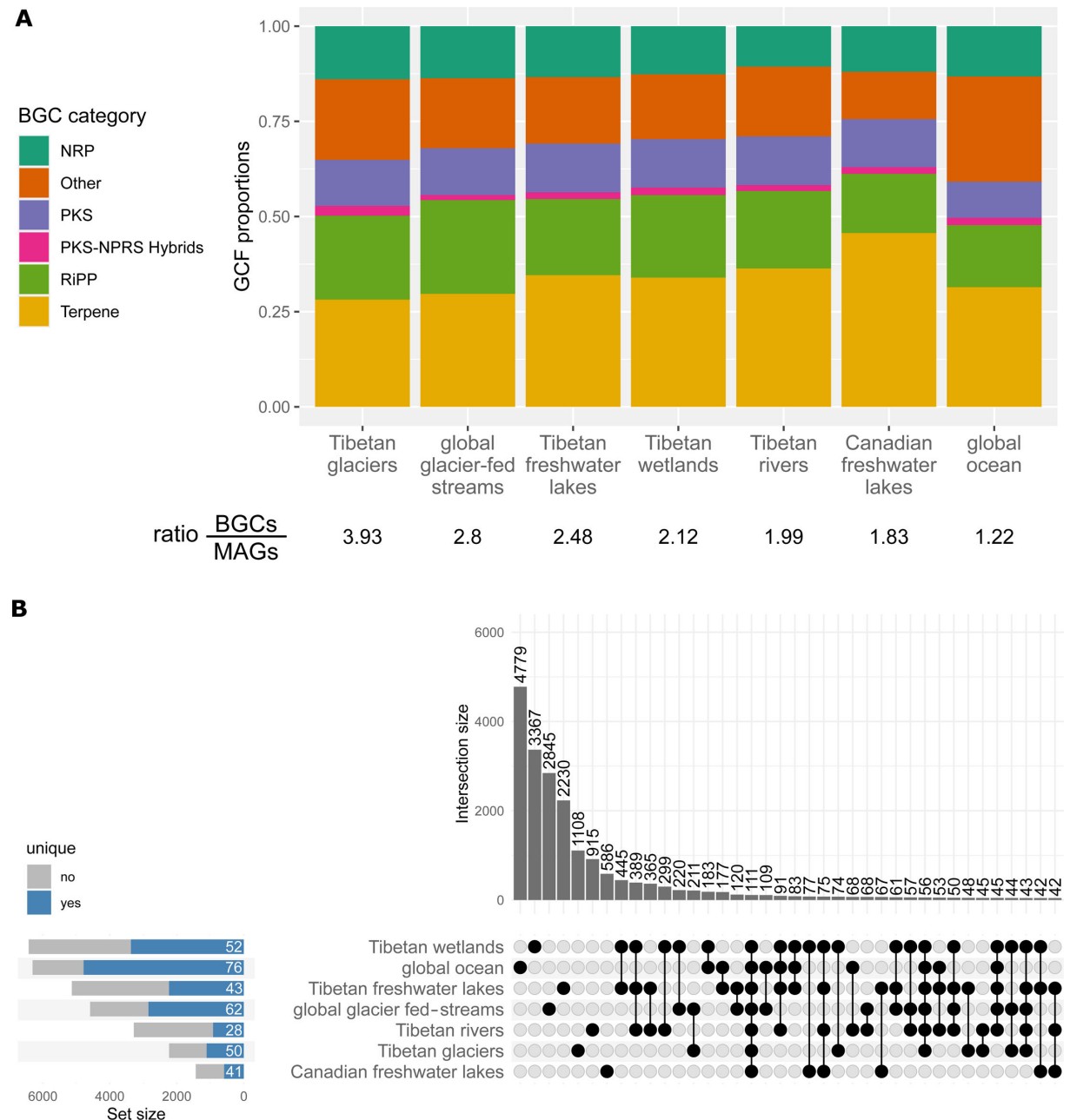

**FIG 2** Comparison of BGC diversity between different aquatic microbiomes and glacier fed streams. (A) Relative prevalence of the BGC categories in the investigated microbiomes. Below the ratio of detected BGCs to MAGs in the respective microbiomes. (B) Overlap of the biosynthetic potential between the different microbiomes (rows) displayed by using an UpSet plot based on GCF presence and absence, where each column depicts an intersection between one or more microbiomes. The bar chart at the top represents the size of the respective intersection, with the exact number of GCFs written at the top. The intersections are ordered by their respective size, and intersections with >40 GCFs are displayed. The bar chart at left side gives the total number of GCFs in the respective microbiomes. GCFs that are only present in one microbiome are marked as unique (blue), and the percentage of unique GCFs per microbiome is written in white.

diversity, we used amplicon sequencing of the 16S and 18S rRNA genes on all samples to determine taxonomic diversity. We found that the Shannon diversity of prokaryotic amplicon sequence variants (ASVs) and GCFs correlated in both epipsammic and epilithic biofilms across all glacier-fed streams (epipsammic: Pearson's r = 0.58, $P$ adj = 8.97 × $10^{-109}$; epilithic: Pearson's r = 0.69, p.adj = 1.67 × $10^{-17}$, Fig. 3C). In contrast, Shannon diversity of photosynthetic eukaryotes (as operational taxonomic units, OTU) correlated

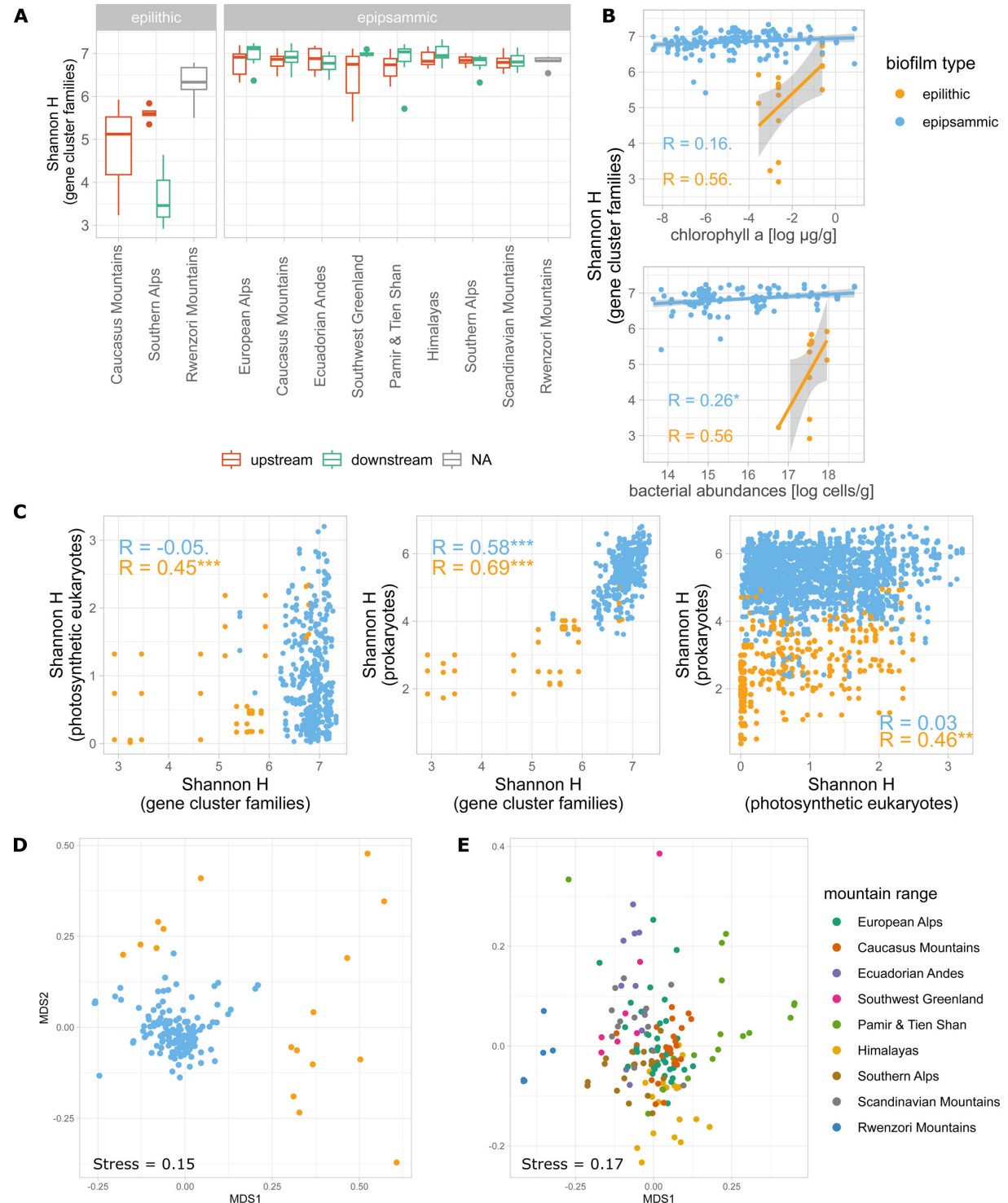

**FIG 3** Diversity of the GCFs in the glacier-fed stream biofilms. (A) Alpha diversity of GCFs in different mountain ranges and epilithic and epipsammic biofilms. (B) Significant correlations of GCF Shannon alpha diversity to chlorophyll *a* and bacterial abundances. (C) Correlations of GCF Shannon alpha diversity to 16S prokaryotic and 18S photosynthetic eukaryotic Shannon diversity. (D) Beta diversity of GCF abundances using an NMDS plot based on Bray–Curtis distances, coloured by biofilm type. (E) Beta diversity of GCF abundances in only the epipsammic biofilms using an NMDS plot based on Bray–Curtis distances, colored by mountain range.

positively with GCF Shannon diversity in epilithic biofilms only (epipsammic: Pearson's r = −0.05, *P* adj = 0.09; epilithic: Pearson's r = 0.45, *P* adj = $6.64 \times 10^{-7}$, Fig. 3C).

Using the relative abundances of GCFs, we found that the composition of GCFs differed significantly between epilithic and epipsammic biofilms (PERMANOVA, $R^2$ = 0.11, $P$ < 0.001, Fig. 3D); this difference is similar to differences in prokaryotic beta-diversity between the two biofilm types (Fig. S2). Additionally, GCF composition in epipsammic biofilms significantly differed between mountain ranges (PERMANOVA $R^2$ = 0.34, $P$ < 0.001, Fig. 3E). Interestingly, we found significant distance–decay patterns of both prokaryotic and GCF beta diversity within most mountain ranges. The prokaryotic microbiome had lower average similarity and a more pronounced distance–decay than GCFs in most mountain ranges (Fig. 4). Distance–decay was most pronounced for both prokaryotic and GCF diversity in the Pamir and Tien Shan mountains (slope = −0.09, Fig. 4), matching the large differences in beta diversity of GCFs (Fig. 3E). Overall, distance–decay of photosynthetic eukaryotes was weak (Fig. 4).

To explore the potential of glacier influence on secondary metabolites, we compared their diversity in epipsammic biofilms from up- and downstream reaches, reflecting the chronosequence of glacier recession (8). While we found lower GCF alpha-diversity in upstream reaches (paired *t*-test, mean difference −0.13, $P$ = 0.0014, Fig. 3A), GCF composition did not significantly differ between reaches across all glacier-fed streams in epipsammic biofilms (PERMANOVA $P$ = 0.15, $R^2$ = 0.009).

## Environmental influences on the biosynthetic potential

To understand potential environmental and biotic controls on the biosynthetic potential, we used differential abundance analysis of GCFs within epipsammic biofilms (Fig. 5). We found that streamwater pH was associated to numerous enriched GCFs, both positively

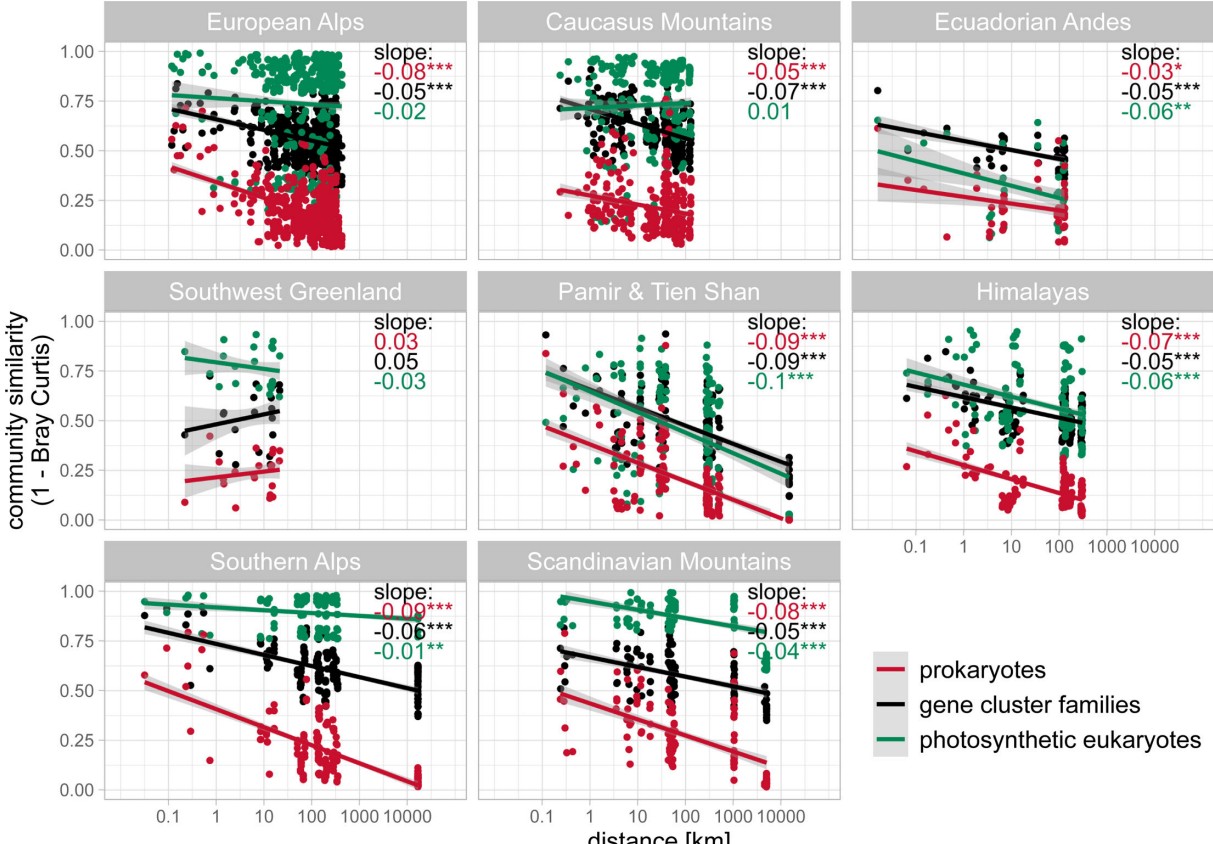

**FIG 4** Distance–decay curves of biosynthetic, prokaryotic, and photosynthetic eukaryote diversity. The respective community similarity (1 − Bray–Curtis dissimilarity) is plotted against the distances between samples. (Epipsammic samples only). The slope of linear model between the community similarity and the distance is displayed in the top right corner, with the adjusted *P*-value of the correlation denoted by stars (*** $P$ < 0.001, **$P$ = 0.001–0.01, *$P$ = 0.01–0.05).

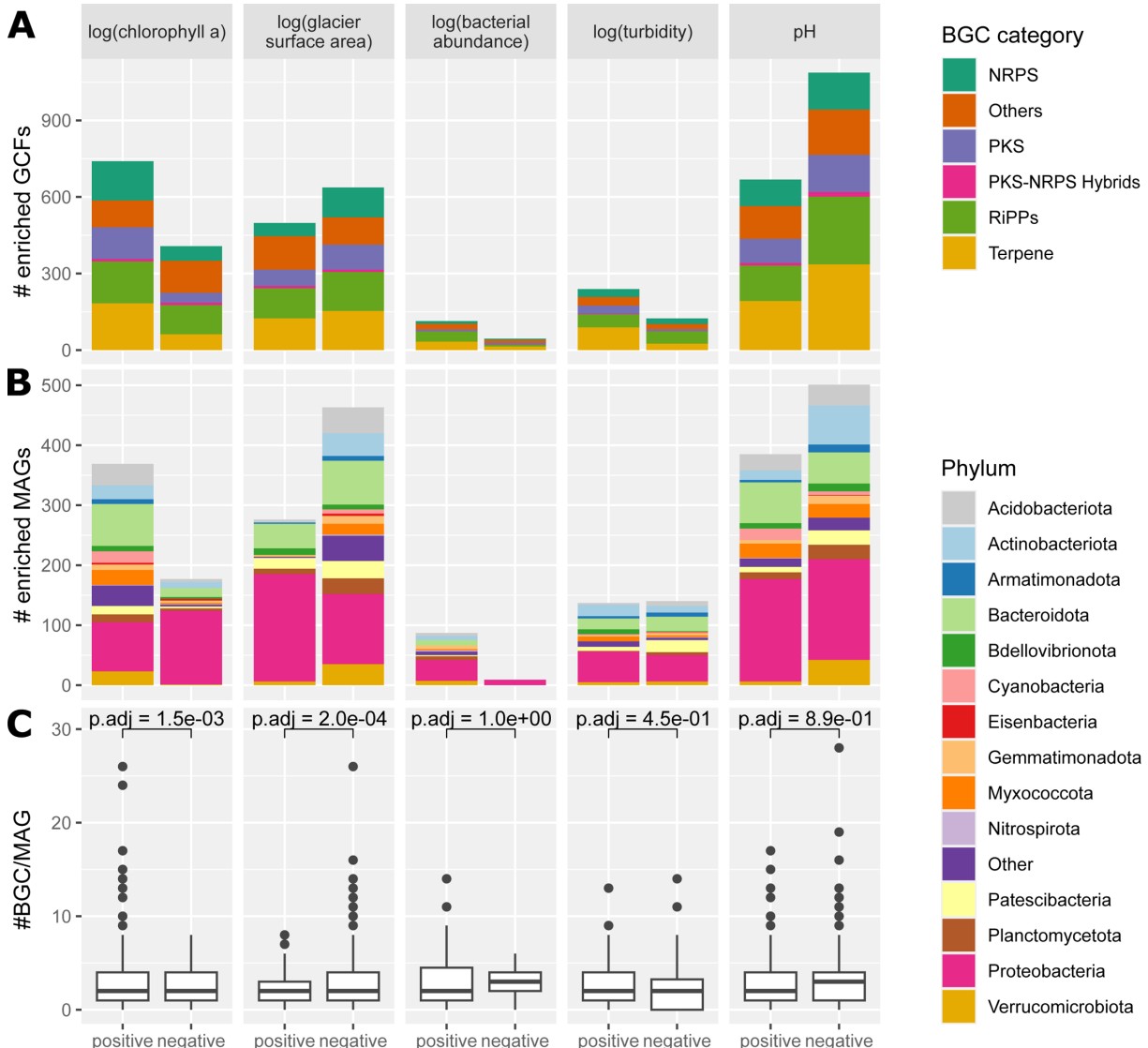

**FIG 5** Investigation of environmental influences on the biosynthetic potential and the microbial community by differential abundance analysis of GCFs (A) and MAGs (B) in epipsammic biofilms for selected environmental variables. Color coded by their BGC category or phylum attribution. Positively associated GCFs/MAGs are always on the left for each environmental variable, and negatively associated are always on the right (same x-axis for all the subplots). (C) Number of BGCs per genome in the differentially enriched MAGs. Testing the difference in the number of BGCs per genome between positively and negatively enriched MAGs resulted in the *P*-values displayed at the top of the bracket (*t*-test, adjusted for multiple testing).

(668) and negatively (1,085); furthermore, 739 GCFs were positively and 407 GCFs were negatively correlated to benthic chlorophyll *a* (Fig. 5A). To further estimate the glacier influence on the microbiome and biosynthetic potential, we investigated GCFs correlated to the glacier surface area, identifying 498 positively and 636 negatively enriched GCFs. Changes in bacterial cell abundance correlated weakly with changes in GCF distribution, as evidenced by 45 negatively enriched and 114 positively enriched GCFs. Differences in composition of the BGC categories between differently enriched groups were minimal (Fig. 5A). These trends were generally reflected in the enrichment of MAGs, and no specific phylum consistently drove these patterns. Still, many MAGs positively correlated to chlorophyll *a* belonged to the phyla Bacteroidota, Cyanobacteria, and Proteobacteria (Fig. 5B). Upon investigating the average number of BGCs per genome, significant differences emerged between positively and negatively enriched MAGs for chlorophyll *a* and glacier surface area (Student's *t*-test *P* adj = $1.5 \times 10^{-3}$, *P* adj

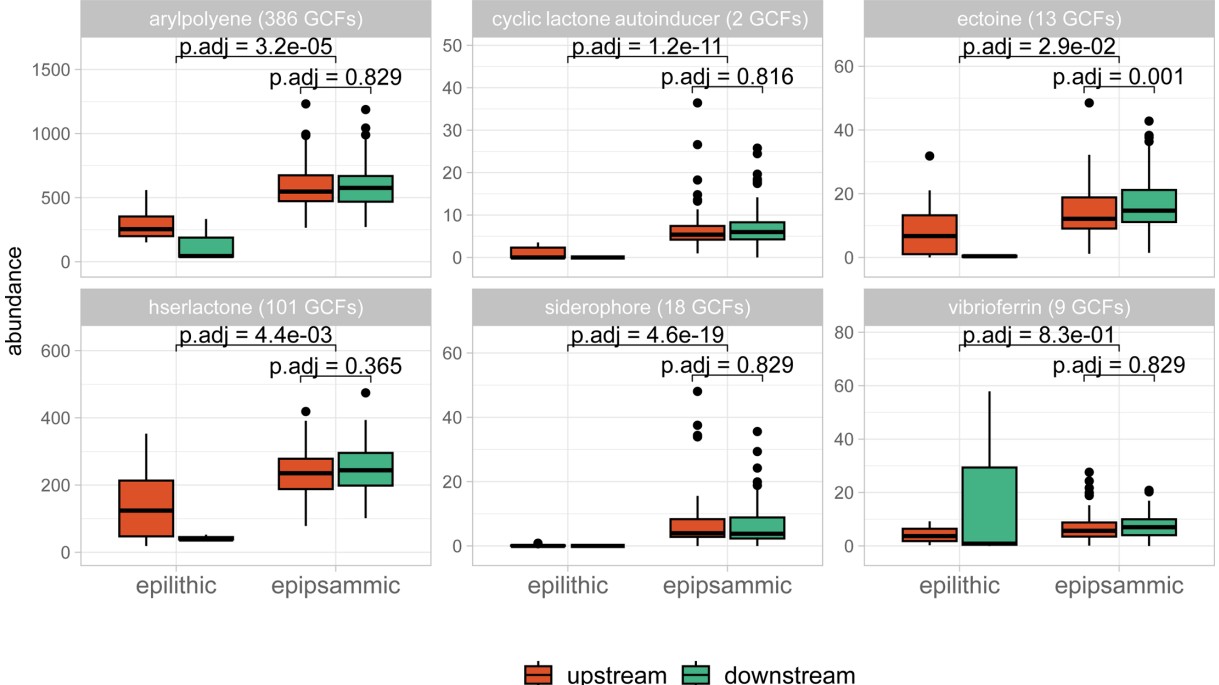

**FIG 6** The abundance of secondary metabolites of interest between epilithic and epipsammic biofilms, and up- and downstream reaches (red and green). In the header in brackets, the number of GCFs per secondary metabolite. Testing the difference in abundance between the epilithic and epipsammic biofilms (*t*-test) and upstream vs downstream epipsammic biofilms (paired *t*-test) resulted in the displayed *P*-values (adjusted for multiple testing).

$= 2.0 \times 10^{-4}$, Fig. 5C). MAGs positively correlated to chlorophyll *a* had an average of 3.18 BGCs per genome, whereas negatively correlated MAGs had a lower average at 2.37 BGCs per genome. MAGs positively correlated to glacier surface area had an average of 2.25 BGCs per genome, while MAGs negatively correlated had higher average of 3 BGCs per genome.

## BGCs encode diverse biological functions

To understand potential roles of BGCs for glacier-fed stream microbiome functioning, we focused on selected secondary metabolites involved in quorum sensing, protection against UV radiation, iron bioavailability, and osmotic and temperature stress. We investigated the difference in abundance between epipsammic and epilithic biofilm and up- and downstream reaches of the secondary metabolites: homo-serine lactones, cyclic lactone autoinducers, arylpolyenes, ectoines, siderophores, and vibrioferrins. We found from two GCFs for cyclic lactone autoinducers up to 386 GCFs for arylpolyenes. No GCFs were directly identified as producing vibrioferrin. However, 12 BGCs were found to group with previously found BGCs resembling vibrioferrin, equalling to nine GCFs (Fig. 6; Table S5). The *recA* normalized abundance of GCFs for the investigated secondary metabolites was higher in epipsammic than epilithic biofilms, resembling patterns of prokaryotic and GCF alpha-diversity (Table S5). Higher abundance in epipsammic biofilms was particularly pronounced for arylpolyenes, cyclic lactone autoinducers, and siderophores, suggesting their importance for epipsammic biofilms. Ectoines were significantly more abundant in epipsammic biofilms from upstream (13.7 ± 6.78) than downstream (17.8 ± 8.72) reaches (paired *t*-test, *P* adj = 0.001) (Fig. 6).

## DISCUSSION

Glacier-fed streams present a unique environment for microorganisms with their limited resources, harsh environmental conditions, and dominance of biofilms. Despite the

unique and endangered nature of this ecosystem, little attention has been given to the biosynthetic potential of the microorganisms dwelling in glacier-fed streams. This study is the first-of-its kind survey on BGCs in glacier-fed stream biofilms around the world. We found a diverse and complex biosynthetic potential, which is distinct from other aquatic microbiomes.

The majority of detected BGCs belong to the category of terpenes and RiPPs, consistent with other metagenomic studies of biosynthetic potential reported from microbiomes from the Greenland Ice Sheet (48), Antarctic soils (40), and pelagic ocean (32). In glacier-fed streams, Myxococcota stood out with the highest average number of BGCs per genome, suggesting a prominent role in signaling and competition within biofilms (Fig. 1A). Myxococcota (Myxobacteria) are known to be prolific producers of secondary metabolites in the ocean (32), activated sludge (49), and glacier ice (39). Their secondary metabolites showed anti-bacterial and anti-viral potential (50), and they are thought to be of relevance for intercellular interactions as predatory weapons (51), complementary to their motility and chemosensory systems (50). Their relatively high coding density (7,400–8,200 genes per 2–7Mb) might also contribute to the high number of BGCs per genome (52). In the laboratory in mono- and co-cultures with prey, Myxobacteria are known to form complex biofilms in which they exchange resources (53). Conversely, they can also inhibit biofilm formation of prey or competitors through secondary metabolites, such as carolacton (50, 54). A phylum which had previously garnered attention for their high BGC diversity are the Eremiobacterota. Marine Eremiobacterota were found to have a high proportion of then novel RiPP BGCs (32). In contrast, we found the Eremiobacterota in glacier-fed streams to have fewer RiPPs and instead higher proportions of terpene BGCs (Fig. 1B), thereby being comparable to Eremiobacterota MAGs from glacier microbiomes (39). This could indicate that these terpenes might be advantageous in cryospheric habitats.

Although at the lower end of the range, the 41% of novel BGCs in glacier-fed streams is comparable to other studies where it ranged from 43% novel BGCs in food fermentations (55), over 56% in marine microbiomes (32), and up to 78% in the Yellow Sea (56). This consistent level of novelty serves to illustrate how much remains to be discovered about BGC diversity in environmental microbiomes. Interestingly, glacier-fed stream Desulfobacterota, which contained many unknown BGCs (Fig. 1C), had not previously been noted as a phylum with high BGC novelty; instead, marine Proteobacteria (56) and Arctic soil Bacteroidota (57) were found to have the most divergent BGCs compared with known secondary metabolites. Elevated BGC novelty within Desulfobacterota could partially be attributed to large proportions of arylpolyene BGCs (Table S2). However, care needs to be taken when comparing the novelty of different BGC categories, as the method is biased to overestimate the similarity of BGCs with few domains (i.e., features), and thus to classify longer BGCs as more novel (36). This might result in a general trend of slightly overestimated novelty for long BGCs (e.g., PKS, NRPS) and slightly underestimated novelty for short BGCs (e.g., terpenes). This is illustrated in our data set by the significant correlation between BGC length and the determined dissimilarity (i.e., distance) to known BGCs (Pearson's r = 0.6, $P = 2.2 \times 10^{-16}$).

Comparing glacier-fed stream microbiome BGCs to other aquatic microbiomes, we found a high percentage of BGCs that are only detected in individual aquatic microbiomes (Fig. 2B). Similarly, high microbiome specificity was observed between closely related microbiomes, such as different food fermentations (55) or different microbiomes on the Greenland Ice Sheet (i.e., ice sheet surface, cryoconite, snow, etc.) (48). High specificity was also observed between more distant microbiomes (e.g., human gut, ocean, and soil) (55). Moreover, this phenomenon can also be observed when comparing the same microbiome between different studies (49). Even for single domains of BGCs (i.e., keto-synthase and adenylation domains of NRPS), high uniqueness between microbiomes, samples, and biological replicates becomes obvious (45, 47, 58). The compendium of specialized metabolite biosynthetic diversity, a collection of BGCs from RefSeq bacterial genomes and MAGs (59), displays a very high biome specificity (~75%)

as well. Important to note here is the higher similarity of BGC content between glacier-fed streams and glaciers than between the other investigated microbiomes (Fig. S1A), suggesting the importance of the environmental conditions in shaping the BGC diversity and composition.

Compared with epilithic biofilms, epipsammic biofilms had a significantly higher BGC diversity (Fig. 3A). Additionally, we found GCFs for several secondary metabolites, which are likely important for biofilms (e.g., quorum sensing molecules, siderophores) to be more abundant in epipsammic biofilms (Fig. 6) independent of *recA* gene abundances (a proxy for cell abundances). However, it is important to take into account the higher prokaryotic and eukaryotic diversity in epipsammic biofilms in glacier-fed streams (44), which may be due to its position in the stream, as well as more and different ecological niches.

Indeed, when interpreting trends in biosynthetic diversity, it is important to consider the positive association with prokaryotic taxonomic diversity (Fig. 3C). This association has previously been observed for the diversity of single domains of BGCs (i.e., keto-synthase and adenylation domains) and bacterial taxonomic diversity across several microbiomes (e.g., soils, marine sediments and marine pelagic water) (45, 60). Additionally, significant correlations between metabolite and taxon richness have been observed (61). Similarly, BGC compositional patterns consistently reflect bacterial taxonomic patterns, as has been observed in marine sediment, seawater, and soil microbiomes (40, 45, 62, 63). We also observed GCF beta diversity patterns reflected in the bacterial beta diversity, differentiating epipsammic and epilithic biofilms (Fig. 3D; Fig. S2). Based on these findings, the biosynthetic diversity is likely related to the microbial diversity in biofilms. This could be a direct causal relationship, whereby a more diverse micro-biome is associated with a more diverse genetic potential. Furthermore, a greater range of secondary metabolites may confer a selective advantage in a complex microbial community or biofilm, by facilitating communication and competition.

Dispersion and environmental conditions can affect secondary metabolite diversity and composition. In the glacier-fed stream microbiomes, we found significant differences in GCF composition between mountain ranges (Fig. 3E). Moreover, for most mountain ranges, there was a significant correlation between distance and GCF dissimilarity (Fig. 4), highlighting the interdependency of geographical distance and biosynthetic composition. We observed that the biosynthetic diversity decreased more slowly over distance in glacier fed stream biofilms, and had also a higher average similarity, than the prokaryotic taxonomic diversity. In soil microbiomes, the composition of single BGC domains (keto-synthase and adenylation domains) show clear distance-dependent trends of dissimilarity, while biome type (i.e., rainforest, temperate forest, desert) has less importance (33). Only on small spatial scales does the biome become more important (33). In marine sediments, on the scale of meters to tens of kilometers, Chase et al. (47) found a significant negative correlation of bacterial community similarity with geographic distance, while the biosynthetic potential showed a much weaker, yet still significant, association to geographic distance (47). A distance–decay pattern in BGCs could be explained by the underlying spatial distribution of the species harboring BGCs and might be watered down by frequent and recent horizontal gene transfer of BGCs (64). A strong selection of BGCs by specific environmental factors, which are spatially structured, might also result in more distinct spatial patterns of BGCs. Another factor to consider is that some secondary metabolites themselves might be either detrimental or beneficial for the dispersion and colonization of microorganisms. Changes in the beta diversity of primary producers could theoretically also lead to changes in biosynthetic potential. However, we did not find significant trends of distance–decay in the photo-synthetic eukaryotes (Fig. 4). Further consideration should be given to methodological influences, such as the taxonomic resolution. For example, using MAG taxonomy yields much shallower distance decay curves, likely because they fail to resolve the microdiver-sity (Fig. S3).

Investigating the association of environmental factors with biosynthetic diversity, we found that a sizeable proportion of GCFs and MAGs was enriched for one or more environmental variables (Fig. 5). We observed many positively and negatively enriched GCFs and MAGs with pH. This suggests an influence of pH on the composition of the microbial community and its biosynthetic potential. This agrees with previous observations in glacier-fed streams where the microbiome was influenced by pH (65). A similar trend was reported from single BGC domains (keto-synthase and adenylation domains) in soil microbiomes (45, 60). It remains to be tested if pH shifts directly affect the biosynthetic potential or if it is a more indirect case where pH drives microbiome composition, which in turn affects the biosynthetic potential.

Furthermore, a higher biosynthetic richness in communities associated with high chlorophyll *a* and reduced glacier influence was observed (Fig. 5). This is in accordance with significantly higher BGC alpha-diversity in downstream reaches (Fig. 3A), which had higher algal biomass. Also, the two ecosystems that favor a biofilm lifestyle, glacier-fed streams and glaciers, displayed the highest overall ratio of BGCs to MAGs (Fig. 2A). Taken together, this could point towards a positive association of secondary metabolite diversity with biofilm formation, overall primary production and chlorophyll *a* concentration. A possible explanation for the positive association could be that there are more intra- and cross-domain interactions in diverse biofilms rich in chlorophyll *a*. These interactions may then favor a higher diversity of secondary metabolites. However, we did not observe a significant correlation between photosynthetic eukaryote alpha diversity and biosynthetic diversity in epipsammic biofilms (Fig. 3C), indicating that it is not the diversity but the abundance of the primary producers that correlates with higher biosynthetic diversity. While the role of secondary metabolites in cross-domain interactions has not been extensively studied, examples, such as siderophore, vibrioferrin, and vitamin B12, demonstrate how eukaryotic algae can profit from secondary metabolites produced by the associated microbiome (19, 66). Previous studies have demonstrated the crucial role of cross-domain interactions, inferred from co-occurrence patterns, in maintaining the stability of the microbiome in glacier-fed streams (6). However, antagonistic interactions might play an important role as well in glacier-fed stream biofilms. A number of antibiotic resistance pathways and BGCs for antibiotics was observed previously, in both pro- and eukaryotes (38). Additionally, a higher abundance of primary producers might correspond to a less energy-limited microbiome, which then further facilitates the energy-demanding biosynthesis of secondary metabolites. In the future a "greening" of glacier-fed streams is expected with decreasing glacier influence (8), including an increase in biofilm and microbial alpha diversity, possibly leading to an increased alpha diversity of secondary metabolite genes.

Based on the high diversity of BGCs in glacier-fed streams, secondary metabolites likely play an important role in the functioning of these microbial communities. However, the culture-independent study of BGCs in MAGs from environmental DNA has several constraints. Due to limited assembly quality and binning of sequences into MAGs, there is a high chance that a number of BGCs are not detected or are fragmented over several contigs, which can lead to an underestimation of the true diversity. Comparing MAGs and whole genome sequences of isolates from the Greenland Ice Sheet shows a clear difference in the ratio of BGCs to genomes, i.e., 7.3 for isolates and 4.8 for MAGs (48). This implies that the biosynthetic richness in glacier-fed streams is likely to be even greater than the estimate presented in this study. Finally, it is also important to note that we only investigated the genetic potential to biosynthesise secondary metabolites. To better understand the role of secondary metabolites in the environments, the expression and synthesis of the actual compounds would need to be investigated as well.

## Conclusion

Our study has established glacier-fed streams as a diverse reservoir of BGCs characterized by numerous novel clusters. Beyond prokaryotic diversity, the unique glacial environment itself shapes BGC composition, potentially influencing the production of secondary

metabolites. Furthermore, the positive association between BGC diversity and biofilm and chlorophyll *a* suggests their critical role within these complex microbial assemblages. With the "greening" of glacier-fed streams, we may see an increased diversity of the genetic potential for secondary metabolites in line with increasing microbial diversity.

## MATERIALS AND METHODS

### Sample collection

Biofilm samples were collected for the *Vanishing Glaciers* project as described previously (7) Briefly, epilithic (i.e., from rocks) and epipsammic (i.e., from fine sediments) biofilms in glacier-fed streams were sampled from nine mountain ranges, including the European Alps, Caucasus Mountains, Ecuadorian Andes, Southwest Greenland, Parim & Tien Shan, Himalayas, Southern Alps, Rwenzori, and Scandinavian Mountains (Table S6). In order to estimate the glacier influence and capture stream heterogeneity, for each stream, a reach was sampled as close as possible to the glacier terminus (up sites), and another reach corresponding with the terminal moraine of the Little Ice Age (if present and distinguishable) was sampled further downstream (down sites). At each stream reach, three independent epipsammic samples were taken by removing the upper 5 cm layer of the streambed and sorting with graded sieves (0.25–3.15 mm size fraction). Meanwhile, epilithic samples were opportunistically (rocks are unequal in their presence due to inherent heterogeneity within and among the streams) taken from the surface of up to three rocks per reach using a metal spatula. All equipment used for sampling was flame-sterilized prior to use. Both epipsammic and epilithic samples were immediately flash-frozen in the field with liquid nitrogen, and samples transported and stored frozen pending analyses. The exception was an aliquot of the epipsammic sample, which was preserved with a paraformaldehyde/glutaraldehyde solution, and reserved for bacterial cell quantification (see below).

Streamwater physicochemical parameters were also measured at each stream reach and included water temperature, pH, specific conductivity, and turbidity—all conducted as previously described (7). Glacier surface area was calculated based on satellite imagery (Sentinel-2; Level 2 a, March 2019–July 2022 from scihub.copernicus.eu) and a catchment definition derived from the ASTER Global Digital Elevation Model (GDEM) v3. (NASA/Meti/Aist/Japan Spacesystems and US/Japan Aster Science Team, 2019). Biomass was estimated two ways for epipsammic (though not epilithic) biofilms in order to estimate the abundance of both photoautotrophs and bacterial cells, respectively. For the former, chlorophyll *a* concentrations were determined by extracting epipsammic biofilms in 90% ethanol for 24 h, followed by fluorescence measurements, and then normalizing resulting values by the dry mass of sediment (7). For the latter, bacterial abundance was quantified by detaching cells from sediments using pyrophosphate and sonication, followed by SybrGreen staining and flow cytometry (NovoCyte, ACEA Biosciences) according to Kohler et al. (67). Finally, DNA was extracted from all available epipsammic and epilithic samples using an established protocol modified specifically for glacier-fed streams, and is based on mechanical bead-beating lysis and phenol:chloroform based extraction and purification (68).

### Amplicon metagenomic sequencing and processing

Amplicon sequencing was performed for each of the three biological replicates, if sufficient biomass was available. For 16S rRNA gene sequencing, the V3–V4 hypervariable regions were amplified using the 341f (5′–528 CCTACGGGNGGCWGCAG-3′) and 785r (5′-GACTACHVGGGTATCTAATCC-3′) 529 primers (69), and libraries prepared as previously described (65). For quantifying eukaryotic diversity, the 18S rRNA gene was amplified using the TAReuk454F (5′-CCAGCA(G/C)C(C/T)GCGGTAATTCC-3′) and TAReukREV3 (5′-AC TTTCGTTCTTGAT(C/T)(A/G)A-3′) (70), and libraries were prepared identically. The libraries

were pooled and sequenced at the Bioscience Cor Lab of the King Abdulla University of Science and Technology, Saudi Arabia, using an Illumina MiSeq 300 bp platform. The resulting reads were processed with the Quantitative Insights Into Microbial Ecology 2 (QIIME2, 2020.8) workflow (71). For the generation of 16S rRNA ASVs, deblur was used, and taxonomy was assigned using the SILVA reference database as previously described (65). Eukaryote, mitochondria, and chloroplast-related sequences were removed. For the computation of prokaryotic Shannon diversity, samples were rarefied to 20,000 reads before computation with the phyloseq package, resulting in 965 samples (Table S4).

The 18S rRNA amplicon reads were similarly processed with QIIME2, and ASVs were generated using DADA2, followed by clustering at 97% similarity to generate OTUs (6). Clustering into OTUs was performed in order to limit possible diversity overestimations due to high copy numbers of 18S rRNA genes in eukaryotes (72). To compute the Shannon diversity of eukaryote photoautotrophs, 18S rRNA OTUs were filtered to include the following phyla and classes: Dinoflagellata, Diatomea, Ochrophyta, Chlorophyta, Prymnesiophyceae, Cryptophyceae, Porphyridiophyceae, Rhodellophyceae, Charophyta, Chlorokybophyceae, Haptophyta, Klebsormidiophyceae, Pavlovophyceae, Florideophycidae, and Perkinsidae. The data set was then rarefied to 10,000 reads, resulting in 968 samples (Table S4).

## Shotgun metagenomic sequencing, processing, assembly, and binning

Shotgun metagenomic libraries were built using 50 ng DNA (from one biological replicate per upstream and downstream reach) by enzymatic fragmentation for 12.5 min and six cycles of PCR amplification according to the protocol of Busi et al. (44), generating 450 bp long DNA fragments. These were sequenced at the Functional Genomics Centre Zurich on a NovaSeq (150 bp, S4 flowcell) platform. The Integrated Meta-omic Pipeline 485 (IMP3) (v3.0) workflow was used to process the reads as previously described (7). Briefly, after adapter trimming, the reads were iteratively assembled with MEGAHIT (v1.2.9) (73). Binning was first performed with Binny (v2.0)(74), MetaBAT2 (v2.12.1) (75), and MAxBin2 (v2.2.7) (76) using single-coverage metagenomic binning. Afterwards, binning was also performed with MetaBAT2 (v2.15), CONCOCT (v1.1.0) (77), and MetaBinner (v1.4.3) (78) with multi-coverage methods relying on the mapped reads of the five spatially-closest samples. DAS Tool (v1.1.4) (79) was used on both set of bins to generate a nonredundant set of bins. All resulting bins with more than 50% contamination were cleaned with MDMCleaner (v0.8.3) (80) and dereplicated at 99% ANI using dRep (v3.2.2) (81). MAGs with a with a minimum completeness of 70% and maximum of 10% contamination, determined by CheckM (v1.0.1) (82), were kept for further analysis. The taxonomy of the MAGs was determined using GTDB-Tk and their coverage estimated using CoverM for 173 samples, which had enough reads (Table S4) (v0.6.1). *RecA* gene (K03553) abundances, for normalization, were similarly estimated by read mapping using CoverM.

## Identification and analysis of BGCs

In order to identify biosynthetic gene clusters, the funcscan pipeline (v1.1.0) (83) was run on the dereplicated prokaryotic MAGs using Nextflow v23.04.1 (84, 85) (Table S1). Within the pipeline, Pyrodigal (v2.1.0) (86) was used to detect open reading frames and AntiSMASH – Lite v6.1.1 (87) was run on contigs > 5 kb to detect BGCs. In a first step, AntiSMASH searches the analyzed sequence against a database of conserved core enzyme HMM profiles, which are indicative of a specific BGC category. In the next step, BGC category specific pre-defined rules (i.e., at least one gene coding for a protein with and AT and KS domain) are employed to identify proto-clusters. These are then extended by its defined neighborhood (i.e., 20 kb) up- and downstream to define the final BGC region. If the core enzymes of a BGC are found close to a contig edge, there might not be enough genes left for the full neighborhood, and the BGC is defined as incomplete. Although AntiSMASH likely misses BGCs with a highly novel structure (because of its rule-based algorithm), it is much more precise than alternative machine-learning based

approaches and still has a relatively high recall (88–90). While eukaryotic MAGs were binned as well, we decided not to investigate BGCs in them since detection sensitivity by AntiSMASH or PlantiSMASH of BGCs in eukaryotic algae has been found to be highly limited (91). To estimate the number of secondary metabolite scaffolds and remedy duplicate counts of BGCs (due to the fragmentation of contigs), all identified BGCs were grouped with BiG-SCAPE (1.1.5) (35) (accessed 2022–11-14) with the mix parameter turned on, a cutoff of 0.3. To assess novelty, glacier-fed stream BGCs were queried against 1.2M pre-processed microbial BGCs from the NCBI database using the query mode in BiG-SLiCE (1.1.1) (36) (threshold 900; Table S1), and the resulting output was further processed using the R package RSQLite (2.3.1) (92) and Tidyverse (2.0.0) (93). The threshold of 900 was previously established by comparing characterized BGCs from MiBiG to each other and indicates that BGCs < 900 synthesize the same secondary metabolite (36). BiG-SLiCE works by converting BGCs into numerical representations. This is achieved by querying BGC gene sequences against a library of profile hidden Markov models (pHMMs) and extracting features based on the presence, absence, and significance of matches. These numerical features are then clustered using a BIRCH algorithm to identify GCFs centroids. Finally, each BGC in the data set or each of the queried BGCs is compared with these GCF models, resulting in a Euclidean distance indicating its membership within each family.

In total, 1797 glacier MAGs (covering snow, ice, and cryoconite habitats with a completeness of >70 a contamination of <10) (39) and 1008 medium and high-quality MAGs from freshwater lake microbiomes (43) were incorporated in the analysis and run through the same funcscan pipeline. The resulting 7,066 glacier and 1,843 lake BGCs, together with 38,232 BGCs from the global ocean microbiome (32) (excluding those from MAGs with <50% completeness and >10% contamination), 44,349 BGCs from Tibetan wetlands, rivers, and lakes (42) (https://zenodo.org/records/5107976), and all the glacier-fed stream BGCs (Table S3) were grouped with BiG-SLiCE (1.1.1) (36) (threshold = 0.4) to generate a full set of GCFs. The ratio of BGCs to MAGs was calculated by taking the total number of detected BGCs or BGCs included in the analysis and dividing it by the total number of MAGs, which were taken as input for the BGC analysis (including the number of MAGs which quality criteria were good enough but which did not have any detected BGCs). Jaccard distances between the investigated microbiomes were calculated with the presence and absence of the GCFs as an input using the vegan R package. The "size" of each GCF was calculated by summing the number of BGCs in each GCFs in each of the microbiomes independently; then, the prevalence of each GCF was calculated by counting the number of microbiomes the GCF was present; and finally size and prevalence were correlated (Fig. S1B).

To estimate the abundance of BGCs in glacier-fed streams, a representative BGC for each GCF was chosen, which was the shortest complete, or if not available, the longest incomplete BGC. This was done in order to more accurately approximate the abundances of the corresponding secondary metabolites and in order to avoid artificially low abundances while read mapping (94). Then, CoverM was run for read mapping on the set of representative BGCs (mean parameter) and 173 samples, which had sufficient sequencing depth (Table S4). The resulting abundances were normalized with the *recA* gene abundances, and instances with a coverage of less than 10% were excluded. Shannon diversity of the GCFs was calculated with the normalized, filtered abundances using the phyloseq R package v1.42.0 (95).

NMDS plots were made using Bray–Curtis distances of the filtered relative abundances of the GCFs. To determine significant clustering, the adonis2 function of the vegan R package (2.6–4) (96) was run on the filtered relative abundances. Distance decay curves of Bray–Curtis dissimilarities were made using the relative abundances of 16S amplicons, 18S amplicons, normalized MAGs, and normalized GCFs.

Differentially abundant GCFs were identified with the MaAsLin2 (microbiome multivariable associations with linear models) R package (1.12.0, method = LM) (97), on the filtered and normalized abundances, for the following environmental variables:

chlorophyll *a*, glacier surface area, bacterial abundance in the sediments, turbidity and pH. Similarly for the identification of differentially abundant MAGs, MaAsLin2 was used on the relative, coverage, and *recA*-normalized MAG abundances. MaAsLin computes linear models for each of the investigated variables, and each of the GCFs and calculates significance values adjusted for multiple testing. Here, we reported GCFs and MAGs as differentially abundant if $P$ adj $<0.05$, denoting them as positively enriched if the coefficient of the model was positive (higher abundance of GCFs correlate with higher values of the variable) and as negatively enriched if the coefficient was negative (higher abundance of the GCF correlates with lower values of the variable). To avoid interferences from sediment types and in order to have enough samples to perform statistically valid analysis, we limited the analysis of differentially abundant GCFs and MAGs to epipsammic biofilm samples.

For the analysis of the selected secondary metabolites, we searched for BGCs labeled as: hserlactones, cyclic lactone autoinducers, arylpolyenes, ectoines, or siderophores, according to BiG-SCAPE product prediction. The summed abundance of all GCFs containing one or more of these BGCs was then used for the analysis and statistical testing. For the analysis of vibrioferrin BGCs, first, the MiBiG accession numbers for vibrioferrin compound were extracted from the database (98). These MIBiG accession numbers and the pre-processed result from BiG-SLiCE on ~1.2M microbial BGCs from the NCBI database were used to identify BGCs resembling vibrioferrin BGCs. In a last step, we identified 12 BGCs (in nine GCFs) in our data set, which showed similarity to these BGCs resembling vibrioferrin BGCs (using the BiG-SLiCE query mode and d < 900).

## ACKNOWLEDGMENTS

The Vanishing Glaciers project is supported by The NOMIS Foundation. Our thanks goes to A. McIntosh and L. Morris in New Zealand, J. Abermann and T. Juul-Pedersen in Greenland, O. Solomina and T. Kuderina Maratovna in Russia, V. Crespo-Pérez and P. Andino Guarderas in Ecuador, J. Yde and S. Leth Jørgensen in Norway, S. Sharma and P. Joshi in Nepal, N. Shaidyldaeva- Myktybekovna and R. Kenzhebaev in Kyrgyzstan, J. Nattabi Kigongo, R. Nalwanga and C. Masembe in Uganda, and M. Gonzlaléz and J. Luis Rodriguez in Chile for logistical support; see https://www.glacierstreams.ch for all institutions involved in the logistics of the expeditions. We acknowledge the help from the porters and guides in Nepal, Uganda, and Kyrgyzstan and E. Oppliger for general laboratory support and the Functional Genomics Centre Zurich for DNA sequencing. This study was also supported by the Charles University project PRIMUS/22/SCI/001 and the Swiss National Science Foundation grant CRSII5_180241.

## AUTHOR AFFILIATIONS

[1]River Ecosystems Laboratory, Alpine and Polar Environmental Research Center, Ecole Polytechnique Fédérale de Lausanne (EPFL), Sion, Switzerland
[2]UK Centre for Ecology and Hydrology (UKCEH), Wallingford, United Kingdom
[3]Department of Ecology, Faculty of Science, Charles University, Prague, Czechia
[4]MARBEC, Univ Montpellier, CNRS, Ifremer, IRD, Montpellier, France

## AUTHOR ORCIDs

Aileen Ute Geers ⓘ http://orcid.org/0000-0002-9162-9423
Grégoire Michoud ⓘ http://orcid.org/0000-0003-1071-9900
Susheel Bhanu Busi ⓘ http://orcid.org/0000-0001-7559-3400
Hannes Peter ⓘ http://orcid.org/0000-0001-9021-3082
Tyler J. Kohler ⓘ http://orcid.org/0000-0001-5137-4844
Leïla Ezzat ⓘ http://orcid.org/0000-0002-4317-6458
Tom J. Battin ⓘ http://orcid.org/0000-0001-5361-2033

## FUNDING

| Funder | Grant(s) | Author(s) |
|---|---|---|
| NOMIS Stiftung (NOMIS Foundation) | | Aileen Ute Geers |
| | | Grégoire Michoud |

## AUTHOR CONTRIBUTIONS

Aileen Ute Geers, Conceptualization, Data curation, Formal analysis, Investigation, Methodology, Visualization, Writing – original draft, Writing – review and editing | Grégoire Michoud, Conceptualization, Data curation, Formal analysis, Methodology, Writing – review and editing | Susheel Bhanu Busi, Conceptualization, Methodology, Writing – review and editing | Hannes Peter, Conceptualization, Supervision, Writing – review and editing | Tyler J. Kohler, Writing – review and editing | Leïla Ezzat, Writing – review and editing | Tom J. Battin, Conceptualization, Funding acquisition, Project administration, Supervision, Writing – review and editing.

## DATA AVAILABILITY

All raw sequencing data and MAGs are available under the bioproject number PRJNA781406.

## ADDITIONAL FILES

The following material is available online.

### Supplemental Material

**Figure S1 (mSystems01137-24-s0001.pdf).**
**Figure S2 (mSystems01137-24-s0002.pdf).** NMDS of prokaryotes.
**Figure S3 (mSystems01137-24-s0003.pdf).** Extended distance decay curves.
**Supplemental material (mSystems01137-24-s0005.docx).** Tables S2 to S5 and legends for other supplemental files.
**Table S1 (mSystems01137-24-s0004.csv).** List of all BGCs found in glacier-fed streams.
**Table S6 (mSystems01137-24-s0006.xlsx).** List of all sampled glacier-fed streams in the *Vanishing Glaciers* project.
**Figure S1 (mSystems01137-24-s0001.pdf).** Jaccard distances between microbiomes based on GCFs and GCF prevalence correlation to GCF size.
**Figure S2 (mSystems01137-24-s0002.pdf).** NMDS of prokaryotes coloured by biofilm type.

### Open Peer Review

**PEER REVIEW HISTORY (review-history.pdf).** An accounting of the reviewer comments and feedback.

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
