## [Reviewer comments · mSystems]

Deciphering the Biosynthetic Landscape of Biofilms in Glacier-Fed Streams

Aileen Geers, Grégoire Michoud, Susheel Busi, Hannes Peter, Tyler Kohler, Leïla Ezzat, and Tom Battin

Corresponding Author(s): Tom Battin, Swiss Federal Institute of Technology

Review Timeline:

Submission Date:	August 23, 2024
Editorial Decision:	October 8, 2024
Revision Received:	November 25, 2024
Accepted:	December 9, 2024

Editor: Hans Bernstein

Reviewer(s): Disclosure of reviewer identity is with reference to reviewer comments included in decision letter(s). The following individuals involved in review of your submission have agreed to reveal their identity: Hannah Doris Schweitzer (Reviewer #1)

Transaction Report:

DOI: <https://doi.org/10.1128/msystems.01137-24>

Re: mSystems01137-24 (Deciphering the Biosynthetic Landscape of Biofilms in Glacier-Fed Streams)

Dear Prof. Tom J Battin:

I apologize that this article has sat in review for longer than I would have liked. One of the reviewers dropped out due to a personal emergency but the other provided (what I consider) a quality review. I have therefore gone through your paper and provided an editorial review in order to keep the process moving in a timely manner. Here are my comments:

Geers et al. present the apparent first large-scale study of biosynthetic gene clusters (BGCs) in glacier-fed stream biofilms, identifying over 8,000 BGCs across 85 streams from major mountain ranges, with more than 40% being novel. Their metagenomic analysis revealed that the BGCs in these biofilms are distinct from those in other aquatic environments, and that BGC diversity is strongly correlated with prokaryotic diversity and biofilm complexity. This study is intended to highlight the unique biosynthetic potential of microorganisms in these changing ecosystems, and the ecological importance of secondary metabolites. As glaciers recede due to climate change, understanding and cataloguing these novel microbial functions becomes increasingly more important.

To be clear, I found little to critique in this paper as the results were well matched with the conclusions, the methods were appropriate, and the delivery was of good quality. Nevertheless, I am providing some detailed points from the editorial bench that I hope can help the authors improve the paper. These changes would help bridge the results and their broader ecological implications, making the study's conclusions clearer and more strongly supported by the data.

More careful clarification on the definition of "novelty" in BGCs (Fig. 1C and related discussion, L167): The concept of novelty is central to the paper, but the explanation of how Euclidean distance ($d > 900$) is used to define novelty in BGCs is unclear. The authors should clarify this threshold in both the text and figure legend, possibly by including more details about the novelty measurement and visualizing the Euclidean distance in a way that makes it easier to interpret how "novel" the BGCs are compared to known BGCs. (more comments below)

The authors can easily provide more context on uniqueness of BGCs across environments (L167-197, Fig. 2): The comparison between glacier-fed streams and other aquatic microbiomes is discussed, but it is not clear how exceptional the novelty of these BGCs is in the broader context. The authors should emphasize whether other aquatic environments typically contain similar proportions of unique BGCs to each other, or if glacier-fed streams are unusually unique. This would strengthen the argument about the distinctive nature of these ecosystems. (also more comments below)

Better explain the correlation between biofilm complexity and BGC diversity (L207-217, Fig. 3): While the correlation between biofilm complexity and BGC diversity is highlighted, the paper could benefit from a more thorough explanation of how this relationship is quantified and what it implies for the functioning of these biofilms. Which environmental/ecological factors have been correlated to BGC diversity before from the other aquatic ecosystems? It would be useful to know up front (maybe even in the Abstract) what is unique about glacial streams' ability to harbor "novel" BGCs.

Similarly... The authors can reinforce the role of environmental factors in shaping BGC diversity (L236-258, Fig. 5): The paper briefly touches on the influence of environmental variables like pH and chlorophyll a on BGC diversity, but this point could be expanded. A deeper exploration of how these factors specifically (mechanistically) drive the biosynthetic potential would make the conclusions/discussion about environmental controls on microbial secondary metabolites more compelling. Additionally, more clearly distinguishing correlation from causation in these sections would strengthen the interpretation of the results (see minor comment below).

The figure legends could be revised to provide a lot more detail that aids the readers. I would suggest directly linking the result that each figure's data supports within the respective legend while also adding more information to help the readers understand the graphs.

The reviewer raised a concern about where the other supporting metagenomic data has been published and under what scientific context. I think more emphasis on this is warranted given the meta-analysis approach to this current study. A geographic map with some comparative details would make a nice figure for this study.

Other comments:

Fig.1: My understanding is that "novelty" is based on the Euclidian distance from known BGCs sourced from the NCBI database. The Fig 1C axis and or figure legend label should reflect this either by explaining in more detail where the threshold of "novel" was derived from or by showing/ranking the Euclidean distances in some other graphical format. It's not exactly clear what "d" is in the notation "($d > 900$)" although I assume it is the distance that I am referring to.

L167, Fig 2 and section: "The biosynthetic potential differs between glacier-fed streams and other aquatic microbiomes". I understand that these results indicate the relative novelty of BCGs from this study to other environments; however, it's not clear from the presentation of results how unexpected this might be. I think I am missing the relative relationship between the other sites to each other. Do each of these contain a similar proportion of unique BSGs from each other or is this something extraordinary to your sample sites? I am also having a hard time reading Fig 2b. More details in the figure legend on how to interpret these graphs in the context of the results communicated are warranted. Should there be labels on the x-axes?

L197: How are these particular biofilms "distinct"?

Fig. 3 and Methods L551-565: I am curious how Shannon's H calculations might be affected by sample size and how this was corrected for uniformly to enable comparisons between taxonomic diversity and BGCs? My understanding is that differences in sample size can create bias and it's not clear how the sample sizes and normalization differs between these two comparators. This might be helpful: <https://doi.org/10.7717/peerj.9391>

Minor comments:

L58: Oddly worded sentence that might want to be revised.

L139: Perhaps it would be useful to the reader to redefine RiPP here? Otherwise they need to back track into the Introduction. Same with PKS, NRPS, and PKS-NPRS. These should also be defined in the Fig. 1 Legend.

L248: Minor terminology issue in this sentence: "Still, many MAGs positively related to chlorophyll a belonged to the phyla Bacteroidota, Cyanobacteria and Proteobacteria (Figure 5B)." This is correlation and not causative relation, correct?

Revision Guidelines

Sincerely,
Hans Bernstein
Editor
mSystems

Reviewer #1 (Comments for the Author):

The authors provided a very large dataset consisting of both metagenome assembled genomes and amplicon data from their own collected benthic glacier-fed stream biofilms and other published works from aquatic microbiomes collected in lakes, wetlands, rivers, and the ocean. The authors focused on the comparison of secondary metabolites, specifically the biosynthetic gene clusters in these environments. The authors identified the genetic potential of these glacier-fed stream biofilm systems suggests that chlorophyll a production and a large diversity of biosynthetic gene clusters play a critical role in the microbial assemblage. The authors also identified that the glacier-fed stream biofilm environment consist of 41% novel BGCs which is consistent with the novelty found in other aquatic microbiomes. Although, the authors did identify previously unidentified BGCs in Desulfobacterota which was previously thought to be a phylum with high BGCs.

This paper is well written and provides in-depth molecular results with valuable statistical tests, geographical patterns and diversity measurements. Although, there are a few areas within the paper that need some clarification for publication. It would be valuable for the authors to include in a table where all the samples come from. What other published works are used and what samples are from this study only? While it is listed in the methods (Line 534-540) what other MAGs are included it would be more valuable to have the list included in Supplementary table 2. While the figures are relevant and well made, the figure legends are lacking leaving it difficult to fully interpret the figure. The legends should be expanded to ensure readers can interpret the figure without the rest of the paper. Also the authors make discussion points that are supported by their results and figures, yet they do not reference any figures throughout the discussion section. The authors should reference those figures when relevant throughout the discussion section and not just the results section.

Minor

Throughout Results and Discussion - There is no need to reference methods whenever you describe something you did.

Remove all "(methods)" references.

Lines 79-84 - The authors give a large list of secondary metabolites and conclude that they might be particularly relevant in glacier fed-stream biofilms but don't explain why they are relevant.

Line 143 - Having this high of an error in your average (2.8+/-2.8) seems like it would be more valuable for the reader to also include the range of the number of BGCs per MAG.

Line 157- 166 - Is the list of BGCs that are novel included somewhere in supplemental? What are the 37 BGCs that have a Euclidean distance >1800?

Line 180-181 - How does Figure 2B show a percentage of uniqueness?

Line 206 - Space at start of paragraph can be removed

Lines 252-256 - These are listed as being statistically significant differences. Do you have a p value or how did you measure the statistical significance?

Lines 311-316 - Can the authors note in a supplementary table which of the BGCs may be overestimated? If the method is flawed is there a way to point out what might be overestimated and which BGCs have more confidence?

Line 328-331 - Noted where? Can the authors reference the figure?

Line 508, 510, and 555 - Is CoverM capitalized or not?

Line 555 - What is the reference? Is it included in the reference list and not numbered here?

Line 570 - Move (Microbiome Multivariable Associations with Linear Models) to the first time MaAsLin2 is mentioned.

Line 580-585 - Why is the analysis of vibrioferrin BGCs the only BGC that has a detailed method for analysis? What about siderophores or arylpolyene or others?

Figure Comments

Figure 1 - How is there a negative number of BGC/MAG. Wouldn't the lowest number of BGCs/MAG be 0?

Figure 2B - This figure is valuable but poorly explained in both the figure legend and the results (see above). The figure legend should explain what the numbers above each bar graph represent.

Figure 1 and 5 - The authors have stacked bar plots with the bottom axis showing the label for all stacked bar plots. This is hard to interpret that the bottom figure is also the axis for all the bar plots. Could the author either make the subfigures closer to each other or better explain in the figure legend that the bottom label is for all the subfigures.

The authors provided a very large dataset consisting of both metagenome assembled genomes and amplicon data from their own collected benthic glacier-fed stream biofilms and other published works from aquatic microbiomes collected in lakes, wetlands, rivers, and the ocean. The authors focused on the comparison of secondary metabolites, specifically the biosynthetic gene clusters in these environments. The authors identified the genetic potential of these glacier-fed stream biofilm systems suggests that chlorophyll a production and a large diversity of biosynthetic gene clusters play a critical role in the microbial assemblage. The authors also identified that the glacier-fed stream biofilm environment consist of 41% novel BGCs which is consistent with the novelty found in other aquatic microbiomes. Although, the authors did identify previously unidentified BGCs in Desulfobacterota which was previously thought to be a phylum with high BGCs.

This paper is well written and provides in-depth molecular results with valuable statistical tests, geographical patterns and diversity measurements. Although, there are a few areas within the paper that need some clarification for publication. It would be valuable for the authors to include in a table where all the samples come from. What other published works are used and what samples are from this study only? While it is listed in the methods (Line 534-540) what other MAGs are included it would be more valuable to have the list included in Supplementary table 2. While the figures are relevant and well made, the figure legends are lacking leaving it difficult to fully interpret the figure. The legends should be expanded to ensure readers can interpret the figure without the rest of the paper. Also the authors make discussion points that are supported by their results and figures, yet they do not reference any figures throughout the discussion section. The authors should reference those figures when relevant throughout the discussion section and not just the results section.

Minor

Throughout Results and Discussion – There is no need to reference methods whenever you describe something you did. Remove all “(methods)” references.

Lines 79-84 – The authors give a large list of secondary metabolites and conclude that they might be particularly relevant in glacier fed-stream biofilms but don't explain why they are relevant.

Line 143 – Having this high of an error in your average (2.8 ± 2.8) seems like it would be more valuable for the reader to also include the range of the number of BGCs per MAG.

Line 157- 166 – Is the list of BGCs that are novel included somewhere in supplemental? What are the 37 BGCs that have a Euclidean distance >1800 ?

Line 180-181 – How does Figure 2B show a percentage of uniqueness?

Line 206 – Space at start of paragraph can be removed

Lines 252-256 – These are listed as being statistically significant differences. Do you have a p value or how did you measure the statistical significance?

Lines 311-316 – Can the authors note in a supplementary table which of the BGCs may be overestimated? If the method is flawed is there a way to point out what might be overestimated and which BGCs have more confidence?

Line 328-331 – Noted where? Can the authors reference the figure?

Line 508, 510, and 555 – Is CoverM capitalized or not?

Line 555 – What is the reference? Is it included in the reference list and not numbered here?

Line 570 – Move (Microbiome Multivariable Associations with Linear Models) to the first time MaAsLin2 is mentioned.

Line 580-585 – Why is the analysis of vibrioferrin BGCs the only BGC that has a detailed method for analysis? What about siderophores or arylpolyene or others?

Figure Comments

Figure 1 – How is there a negative number of BGC/MAG. Wouldn't the lowest number of BGCs/MAG be 0?

Figure 2B – This figure is valuable but poorly explained in both the figure legend and the results (see above). The figure legend should explain what the numbers above each bar graph represent.

Figure 1 and 5 – The authors have stacked bar plots with the bottom axis showing the label for all stacked bar plots. This is hard to interpret that the bottom figure is also the axis for all the bar plots. Could the author either make the subfigures closer to each other or better explain in the figure legend that the bottom label is for all the subfigures.

Response to reviewers

Editor comments:

Geers et al. present the apparent first large-scale study of biosynthetic gene clusters (BGCs) in glacier-fed stream biofilms, identifying over 8,000 BGCs across 85 streams from major mountain ranges, with more than 40% being novel. Their metagenomic analysis revealed that the BGCs in these biofilms are distinct from those in other aquatic environments, and that BGC diversity is strongly correlated with prokaryotic diversity and biofilm complexity. This study is intended to highlight the unique biosynthetic potential of microorganisms in these changing ecosystems, and the ecological importance of secondary metabolites. As glaciers recede due to climate change, understanding and cataloguing these novel microbial functions becomes increasingly more important.

To be clear, I found little to critique in this paper as the results were well matched with the conclusions, the methods were appropriate, and the delivery was of good quality. Nevertheless, I am providing some detailed points from the editorial bench that I hope can help the authors improve the paper. These changes would help bridge the results and their broader ecological implications, making the study's conclusions clearer and more strongly supported by the data.

We are grateful for this thoughtful and constructive feedback on our manuscript. We have carefully considered each point raised and have made appropriate revisions to the manuscript to address the suggestions. We have revised the manuscript to provide a more precise explanation of novelty and uniqueness, and to include more comprehensive figure legends. We believe that these revisions have substantially improved the clarity and impact of our manuscript.

More careful clarification on the definition of "novelty" in BGCs (Fig. 1C and related discussion, L167): The concept of novelty is central to the paper, but the explanation of how Euclidean distance ($d > 900$) is used to define novelty in BGCs is unclear. The authors should clarify this threshold in both the text and figure legend, possibly by including more details about the novelty measurement and visualizing the Euclidean distance in a way that makes it easier to interpret how "novel" the BGCs are compared to known BGCs. (more comments below)

Indeed, the determination of BGC novelty can be difficult and hence requires more careful explanation. In this study we employed the program BiG-SLiCE to compare glacier-fed stream BGCs to a collection of 1.2 million BGCs from NCBI. BiG-SLiCE extracts PFAM domains from the BGCs and uses the presence and absence of these PFAMs to compute the Euclidean distances of the BGCs to each other. The threshold of a Euclidean distance > 900 of BGC to BGCs in the BiG-SLiCE database has been previously determined using a set of characterised BGCs (<https://academic.oup.com/gigascience/article/10/1/giaa154/6092777>) and indicates that BGCs with $d > 900$ are likely to produce different secondary metabolites to the known ones. We clarified this in the revised manuscript by adding a more detailed explanation of how novelty was determined in the results (lines 164-170) and in the Figure 1 legend. Additionally, we further clarified the threshold in the methods (lines 558-566).

The authors can easily provide more context on uniqueness of BGCs across environments (L167-197, Fig. 2): The comparison between glacier-fed streams and other aquatic microbiomes is discussed, but it is not clear how exceptional the novelty of these BGCs is in the broader context. The authors should emphasize whether other aquatic environments typically contain similar proportions of

unique BGCs to each other, or if glacier-fed streams are unusually unique. This would strengthen the argument about the distinctive nature of these ecosystems. (also more comments below)

We agree, this aspect of the manuscript required further attention. While our analysis indicated that the proportion of uniqueness ranged between 28% and 76% in river, glacier, lakes and ocean microbiomes, 62% of the gene cluster families were uniquely detected in glacier-fed streams. Hence, glacier-fed streams indeed do not harbour an extraordinarily high degree of uniqueness, but high uniqueness and biome specificity seem to be a common feature for BGCs and have been observed before in other environments, including soils (see discussion (lines: 333-342)).

In order to better highlight the proportion of uniqueness in glacier fed streams, we added more information in the results section and explained the Figure 2B in more detail (lines: 189-195). Additionally, we highlighted the proportion of unique GCFs in Figure 2B.

Better explain the correlation between biofilm complexity and BGC diversity (L207-217, Fig. 3): While the correlation between biofilm complexity and BGC diversity is highlighted, the paper could benefit from a more thorough explanation of how this relationship is quantified and what it implies for the functioning of these biofilms.

We assume that based on the lines mentioned you are referring specifically to the prokaryotic and phototrophic eukaryotic diversity in this context. Indeed, prokaryotic diversity is strongly correlated to BGC diversity in glacier-fed streams. This might be a direct causative relationship or/and a more diverse biosynthetic potential might be favourable in a complex microbiome/biofilm. We expanded on this further in the discussion section (lines 362-367).

Which environmental/ecological factors have been correlated to BGC diversity before from the other aquatic ecosystems?

To date little is known about how ecological and environmental factors are associated with BGC diversity. A few studies have identified taxonomic diversity as a correlated factor. Specifically for the other aquatic ecosystems we know very little. The studies focusing on the Tibetan aquatic and glacier microbiomes did not investigate associations to environmental factors. In the global ocean, differences in the biosynthetic potential between different sample types (prokaryote vs virus vs particle enriched, different latitudes, water depth, temperature) have been observed, but no correlations of the BGC diversity to environmental factors were performed.

It would be useful to know up front (maybe even in the Abstract) what is unique about glacial streams' ability to harbor "novel" BGCs.

We agree that glacier-fed streams represent a unique environment and present an opportunity to study a unique set of secondary metabolites. In particular, previous work has shown that the permanently cold, ultra-oligotrophic and physically unstable glacier-fed stream environment selects for a unique benthic biofilm microbiome, dominated by a few but well-adapted taxa. We now better highlight these aspects in relation to BGCs in the abstract (lines 23-39).

Similarly... The authors can reinforce the role of environmental factors in shaping BGC diversity (L236-258, Fig. 5): The paper briefly touches on the influence of environmental variables like pH and chlorophyll a on BGC diversity, but this point could be expanded. A deeper exploration of how these factors specifically (mechanistically) drive the biosynthetic potential would make the

conclusions/discussion about environmental controls on microbial secondary metabolites more compelling.

While we can be relatively certain that changes in environmental factors are associated with changes in the biosynthetic potential, a mechanistic understanding of how, for instance, pH may shape BGC diversity would require experimental validation, which is beyond the scope of this work. However, we agree that a deeper exploration of potential mechanism at work strengthens the manuscript. .

For chlorophyll a, we hypothesized that a higher chlorophyll a concentration correlates with an increase in intra- and cross domain interactions, which would be reflected by an increased diversity of secondary metabolites. We further clarified and highlighted this in the discussion (lines: 404-413). Additionally, the energetic potential produced by the primary producers in the microbiome, might facilitate the energy-demanding biosynthesis of secondary metabolites (lines 425-427).

For pH it is more difficult to tell what mechanistic drivers might result in changes of the biosynthetic potential. Also, important to note for pH is that changes in the biosynthetic potential might directly result from environmental changes or alternatively more indirectly result from microbiome changes, which we further highlighted in the discussion (lines: 401-403).

Additionally, more clearly distinguishing correlation from causation in these sections would strengthen the interpretation of the results (see minor comment below).

We agree and clarify that as this work represents a purely observational study, our observations are correlative. To clarify, we rephrased several sentences in the results section (lines: 249-268).

The figure legends could be revised to provide a lot more detail that aids the readers. I would suggest directly linking the result that each figure's data supports within the respective legend while also adding more information to help the readers understand the graphs.

Thank you for the valuable input, as suggested by you and the reviewer we explained the figures in more detail in the legend and linked them to the result sections.

The reviewer raised a concern about where the other supporting metagenomic data has been published and under what scientific context. I think more emphasis on this is warranted given the meta-analysis approach to this current study. A geographic map with some comparative details would make a nice figure for this study.

To clarify where this data is coming from, we added a supplementary table listing all MAGs and BGCs included in this study with their respective references to the different studies (revised Supplementary table 3). However, MAGs (and their BGCs) identified in other studies may occur in several samples (e.g. the LakePulse consortium sampled 308 lakes) and information about the spatial distribution of these MAGs is unfortunately not always available, precluding an accurate map of the spatial origin of these MAGs.

For the geographic locations of glacier-fed streams we added another supplementary table (revised Supplementary table 6) listing the sampled glacier-fed streams and their elevations and coordinates. Additionally a map of the sampled glacier-fed streams can be found in the paper of Michoud et al. (Michoud G, Peter H, Busi SB, Bourquin M, Kohler TJ, Geers A, Ezzat L, The Vanishing Glaciers Field Team, Battin TJ. Mapping the metagenomic diversity of the multi-kingdom glacier-fed stream microbiome. Submitted.)

Other comments:

Fig.1: My understanding is that "novelty" is based on the Euclidian distance from known BGCs sourced from the NCBI database. The Fig 1C axis and or figure legend label should reflect this either by explaining in more detail where the threshold of "novel" was derived from or by showing/ranking the Euclidean distances in some other graphical format. It's not exactly clear what "d" is in the notation "(d > 900)" although I assume it is the distance that I am referring to.

Yes, this is correct. We expanded the figure legend accordingly and also further clarified novelty in the main text (lines: 164-170).

L167, Fig 2 and section: "The biosynthetic potential differs between glacier-fed streams and other aquatic microbiomes". I understand that these results indicate the relative novelty of BGCs from this study to other environments; however, it's not clear from the presentation of results how unexpected this might be. I think I am missing the relative relationship between the other sites to each other. Do each of these contain a similar proportion of unique BSGs from each other or is this something extraordinary to your sample sites? I am also having a hard time reading Fig 2b. More details in the figure legend on how to interpret these graphs in the context of the results communicated are warranted. Should there be labels on the x-axes?

We expanded the Figure legend to better explain how this plot is structured. Additionally, we further highlighted the percentage of unique BGCs in the individual environments in the figure itself by colouring them in in blue. This makes it clearer that all of the microbiomes have a relatively large proportion of unique BGCs. So, while glacier-fed streams display a high uniqueness it is not extraordinary in comparison to the other environments.

L197: How are these particular biofilms "distinct"?

We rephrased the sentence to make it clearer that the epipsammic biofilm community is distinct to the epilithic biofilm community.

Fig. 3 and Methods L551-565: I am curious how Shannon's H calculations might be affected by sample size and how this was corrected for uniformly to enable comparisons between taxonomic diversity and BGCs? My understanding is that differences in sample size can create bias and it's not clear how the sample sizes and normalization differs between these two comparators. This might be helpful: <https://doi.org/10.7717/peerj.9391>

Differences in sample size can indeed create biases for Shannon calculations. In order to compensate for that, we rarefied the amplicon reads to estimate taxonomic diversity (lines: 502-504, 513-514). The GCF abundances were normalized by the abundances of the recA gene, which also compensates for differences in metagenome sequencing depth (lines 591-592). Of course, both rarefaction and recA gene normalization are not perfect, but we still believe the final Shannon estimates are good enough estimates. In particular for seeing the general trends for correlations and distance decay trends.

Minor comments:

L58: Oddly worded sentence that might want to be revised.

We revised the sentence to make it clearer.

L139: Perhaps it would be useful to the reader to redefine RiPP here? Otherwise they need to back track into the Introduction. Same with PKS, NRPS, and PKS-NPRS. These should also be defined in the Fig. 1 Legend.

As suggested, we redefined the different BGC categories in the results section and figure 1 legend.

L248: Minor terminology issue in this sentence: "Still, many MAGs positively related to chlorophyll a belonged to the phyla Bacteroidota, Cyanobacteria and Proteobacteria (Figure 5B)." This is correlation and not causative relation, correct?

Yes, there are no causative effects. We rephrased the sentence for clarification.

Reviewer #1 (Comments for the Author):

The authors provided a very large dataset consisting of both metagenome assembled genomes and amplicon data from their own collected benthic glacier-fed stream biofilms and other published works from aquatic microbiomes collected in lakes, wetlands, rivers, and the ocean. The authors focused on the comparison of secondary metabolites, specifically the biosynthetic gene clusters in these environments. The authors identified the genetic potential of these glacier-fed stream biofilm systems suggests that chlorophyll a production and a large diversity of biosynthetic gene clusters play a critical role in the microbial assemblage. The authors also identified that the glacier-fed stream biofilm environment consist of 41% novel BGCs which is consistent with the novelty found in other aquatic microbiomes. Although, the authors did identify previously unidentified BGCs in Desulfobacterota which was previously thought to be a phylum with high BGCs.

This paper is well written and provides in-depth molecular results with valuable statistical tests, geographical patterns and diversity measurements. Although, there are a few areas within the paper that need some clarification for publication.

We thank the reviewer for their positive assessment of our work and their helpful suggestions for improvement. We have carefully considered the reviewer's comments and have made revisions to the manuscript to address the points raised.

It would be valuable for the authors to include in a table where all the samples come from. What other published works are used and what samples are from this study only? While it is listed in the methods (Line 534-540) what other MAGs are included it would be more valuable to have the list included in Supplementary table 2.

Supplementary table 2 (revised number 4) lists samples from the Vanishing Glaciers Project and we included references to the other studies, in which samples from the Vanishing Glaciers Project were analysed in the legend. To better clarify which data from other work was included, we included an additional supplementary table (revised Supplementary table 3) with the number of MAGs and BGCs included. We did not combine those two tables because they list different data types, one does have a list of samples from which abundances can be estimated, the other is a list of just biological entities. Please see also our reply to similar concerns raised by the editor.

While the figures are relevant and well made, the figure legends are lacking leaving it difficult to fully interpret the figure. The legends should be expanded to ensure readers can interpret the figure without the rest of the paper.

Thank you for the feedback on the figures, we carefully expanded several figure legends to make them more accessible.

Also the authors make discussion points that are supported by their results and figures, yet they do not reference any figures throughout the discussion section. The authors should reference those figures when relevant throughout the discussion section and not just the results section.

We agree and added references to the figures in the discussion as well.

Minor

Throughout Results and Discussion - There is no need to reference methods whenever you describe something you did. Remove all "(methods)" references.

As suggested, we removed all "(methods)" references.

Lines 79-84 - The authors give a large list of secondary metabolites and conclude that they might be particularly relevant in glacier fed-stream biofilms but don't explain why they are relevant.

We have chosen to investigate these specific secondary metabolites because of the unique environmental conditions in glacier-fed streams and the dominant biofilm mode of life. We included this reasoning in the introduction now as well (lines: 82-88).

Line 143 - Having this high of an error in your average (2.8 ± 2.8) seems like it would be more valuable for the reader to also include the range of the number of BGCs per MAG.

As suggested, we included the range of BGCs per MAG as well.

Line 157- 166 - Is the list of BGCs that are novel included somewhere in supplemental? What are the 37 BGCs that have a Euclidean distance >1800 ?

To provide a better overview of the BGCs found in glacier-fed streams and their novelty we added a supplementary table listing all BGCs and their Euclidean distances to known BGCs (revised Supplementary table 1).

Line 180-181 - How does Figure 2B show a percentage of uniqueness?

The old version of the figure does not show the percentage directly. However, one can compare the complete set size of the individual microbiomes to their respective intersection size (i.e. the number of GCFs only present in one specific microbiome) and estimate what the proportion of uniqueness is. We expanded on the figure legend and removed the reference to the Figure in the text to make it clearer. We also modified the figure 2B to highlight the uniqueness in blue in the left side bars and added the percentage of uniqueness written in white in the bars.

Line 206 - Space at start of paragraph can be removed

As pointed out, we removed the space at the start of the paragraph.

Lines 252-256 - These are listed as being statistically significant differences. Do you have a p value or how did you measure the statistical significance?

We compared the average number of BGCs per genome and tested the significance using a student's t-test. The adjusted p-value for this can be found on lines 263-264 in the revised manuscript. Lines 311-316 - Can the authors note in a supplementary table which of the BGCs may be overestimated? If the method is flawed is there a way to point out what might be overestimated and which BGCs have more confidence?

No, it is not possible to point out specific BGCs, for which their novelty is over- or underestimated. In general there is a trend that the novelty of BGCs with few domains, such as terpenes, might be underestimated, and that the novelty of BGCs with a lot of domains, such as large NRPS or PKS, might be overestimated. We expanded on that a bit further in the main text (lines: 326-328).

Line 328-331 - Noted where? Can the authors reference the figure?

This can be seen in the Jaccard distances between the environments displayed in supplementary figure 1A. We added the reference in the main text.

Line 508, 510, and 555 - Is CoverM capitalized or not?

Yes, it is. We adjusted accordingly throughout the text.

Line 555 - What is the reference? Is it included in the reference list and not numbered here?

We are sorry, it was a forgotten reference. We corrected it to be included in the reference list.

Line 570 - Move (Microbiome Multivariable Associations with Linear Models) to the first time MaAsLin2 is mentioned.

We moved the full spelling of MaAsLin2 to the first time it is mentioned.

Line 580-585 - Why is the analysis of vibrioferrin BGCs the only BGC that has a detailed method for analysis? What about siderophores or arylpolyene or others?

The analysis of vibrioferrin BGCs was specifically listed because it was a special case in contrast to the other BGCs. We added more information on how the other BGCs were analysed (lines: 614-617).

Figure Comments

Figure 1 - How is there a negative number of BGC/MAG. Wouldn't the lowest number of BGCs/MAG be 0?

Yes, you are right the lowest number is 0. There are no negative numbers of BGCs/MAG. Maybe the black horizontal line denoting average number of BGCs/MAG across all phyla got confused for the 0 line? We changed the colour of this line to blue to make it clearer.

Figure 2B - This figure is valuable but poorly explained in both the figure legend and the results (see above). The figure legend should explain what the numbers above each bar graph represent.

We expanded the figure legend and added additional text in result section (lines: 189-195) for a better explanation.

Figure 1 and 5 - The authors have stacked bar plots with the bottom axis showing the label for all stacked bar plots. This is hard to interpret that the bottom figure is also the axis for all the bar plots.

Could the author either make the subfigures closer to each other or better explain in the figure legend that the bottom label is for all the subfigures.

We added clarification to the figure legends and moved the subfigures closer to each other.

Re: mSystems01137-24R1 (Deciphering the Biosynthetic Landscape of Biofilms in Glacier-Fed Streams)

Dear Prof. Tom J Battin:

I reviewed this revised version and communicated with the reviewer, and this new version addressed all of the concerns raised.

Your manuscript has been accepted, and I am forwarding it to the ASM production staff for publication. Your paper will first be checked to make sure all elements meet the technical requirements. ASM staff will contact you if anything needs to be revised before copyediting and production can begin. Otherwise, you will be notified when your proofs are ready to be viewed.

Sincerely,

Hans Bernstein
Editor
mSystems